# Learning without the Phase: Regularized PhaseMax Achieves Optimal Sample Complexity

**Fariborz Salehi**
Department of Electrical Engineering
Caltech
fsalehi@caltech.edu

**Ehsan Abbasi**
Department of Electrical Engineering
Caltech
eabbasi@caltech.edu

**Babak Hassibi**
Department of Electrical Engineering
Caltech
hassibi@caltech.edu[*]

## Abstract

The problem of estimating an unknown signal, $\mathbf{x}_0 \in \mathbb{R}^n$, from a vector $\mathbf{y} \in \mathbb{R}^m$ consisting of $m$ magnitude-only measurements of the form $y_i = |\mathbf{a}_i \mathbf{x}_0|$, where $\mathbf{a}_i$'s are the rows of a known measurement matrix $\mathbf{A}$, is a classical problem known as phase retrieval. This problem arises when measuring the phase is costly or altogether infeasible. In many applications in machine learning, signal processing, statistics, etc., the underlying signal has certain structure (sparse, low-rank, finite alphabet, etc.), opening of up the possibility of recovering $\mathbf{x}_0$ from a number of measurements smaller than the ambient dimension, i.e., $m < n$. Ideally, one would like to recover the signal from a number of phaseless measurements that is on the order of the "degrees of freedom" of the structured $\mathbf{x}_0$. To this end, inspired by the PhaseMax algorithm, we formulate a convex optimization problem, where the objective function relies on an initial estimate of the true signal and also includes an additive regularization term to encourage structure. The new formulation is referred to as **regularized PhaseMax**. We analyze the performance of regularized PhaseMax to find the minimum number of phaseless measurements required for perfect signal recovery. The results are asymptotic and are in terms of the geometrical properties (such as the Gaussian width) of certain convex cones. When the measurement matrix has i.i.d. Gaussian entries, we show that our proposed method is indeed order-wise optimal, allowing perfect recovery from a number of phaseless measurements that is only a constant factor away from the optimal number of measurements required when phase information is available. We explicitly compute this constant factor, in terms of the quality of the initial estimate, by deriving the exact phase transition. The theory well matches empirical results from numerical simulations.

## 1 Introduction

Recovering an unknown signal or model given a limited number of linear measurements is an important problem that appears in many applications. Researchers have developed various methods

---

[*]This work was supported in part by the National Science Foundation under grants CNS-0932428, CCF-1018927, CCF-1423663 and CCF-1409204, by a grant from Qualcomm Inc., by NASA's Jet Propulsion Laboratory through the President and Director's Fund, and by King Abdullah University of Science and Technology.

with rigorous theoretical guarantees for perfect signal reconstruction, e.g. [5, 16, 38, 43]. However, there are many practical scenarios in which the signal should be reconstructed from nonlinear measurements. In particular, in many physical devices, measuring the phase is expensive or even infeasible. For instance, detection devices such as CCD cameras and photosensitive films cannot measure the phase of a light wave and instead measure the photon flux [22].

The fundamental problem of recovering a signal from magnitude-only measurements is known as *phase retrieval*. It has a rich history and occurs in many areas in engineering and applied sciences such as medical imaging [15], X-ray crystallography [27], astronomical imaging [17], and optics [45]. Due to the loss of phase information, signal reconstruction from magnitude-only measurements can be quite challenging. Therefore, despite a variety of proposed methods and analysis frameworks, phase retrieval still faces fundamental theoretical and algorithmic challenges.

Recently, convex methods have gained significant attention to solve the phase retrieval problem. The first convex-relaxation-based methods were based on semidefinite programs [7, 10] and resorted to the idea of *lifting* [2, 8, 23, 36] the signal from a vector to a matrix to linearize the quadratic constraints. While the convex nature of this formulation allows theoretical guarantees, the resulting algorithms are computationally inefficient since the number of unknowns is effectively squared. This makes these approaches intractable when the system dimension is large.

Introduced in two independent papers [3, 19], PhaseMax is a novel convex relaxation for phase retrieval which works in the original $n$-dimensional parameter space. Since it does not require lifting and does not square the number of unknowns, it is appealing in practice. It does, however, require an intial estimate of the signal. Preliminary theoretical analysis [3, 13, 19, 21] indicates the method achieves perfect recovery for an order optimal number of random measurements. The exact phase transition for PhaseMax has been recently computed in a sequence of papers, first for the case of real measurements [14] and then for the case of complex ones [35].

Non-convex methods for phase retrieval have a long history [18]. Recent non-convex methods start with a careful initialization [25, 26, 28] and update the solution iteratively using a gradient-descent-like scheme. Examples of such methods include Wirtinger flow algorithms [9, 12, 37], truncated amplitude flow [46], and alternating minimization [29, 49]. Despite having lower computational cost, precise theoretical analysis of such algorithms seems very technically challenging.

All the aforementioned algorithms essentially demonstrate that a signal of dimension $n$ can be perfectly recovered through $m > Cn$ amplitude-only measurements, where $C > 1$ is a constant that depends on the algorithm as well as the measurement vectors. However, many interesting signals in practice contain fewer degrees of freedom than the ambient dimension (sparse signals, low-rank matrices, finite alphabet signals, etc.). Such low-dimensional structures open up the possibility of perfect signal recovery with a number of measurements significantly smaller than $n$.

## 1.1   Summary of contributions

In this paper we propose a new approach for recovering *structured* signals. Inspired by the PhaseMax algorithm, we introduce a new convex formulation and investigate necessary and sufficient conditions, in terms of the number of measurements, for perfect recovery. We refer to this new framework as *regularized PhaseMax*. The constrained set in this optimization is obtained by relaxing the non-convex equality constraints in the original phase retrieval problem to convex inequality constraints. The objective function consists of two terms. One is a linear functional that relies on an initial estimate of the true signal which must be externally provided. The second term is an additive regularization term that is formed based on a priori structural information about the signal.

We utilize the recently developed Convex Gaussian Min-Max Theorem (CGMT) [39] to precisely compute the necessary and sufficient number of measurements for perfect signal recovery when the entries of the measurement matrix are i.i.d. Gaussian. To the extent of our knowledge, this is the first convex optimization formulation for the problem of structured signal recovery given phaseless linear Gaussian measurements that provably requires an order optimal number of measurements. In this paper we focus on real signals and real measurements. The complex case is more involved, requires a different analysis, and will be considered in a separate work. Through our analysis, we make the following main contributions:

- We first provide a sufficient recovery condition, in Section 3.1, in terms of the number of measurements, for perfect signal recovery. We use this to infer that our proposed method is order-wise optimal.

- We characterize the exact phase transition behavior for the class of absolutely scalable regularization functions.

- We apply our findings to two special examples: unstructured signal recovery and sparse recovery. We observe that the theory well matches the result of numerical simulations for these two examples.

## 1.2 Prior work

Phase retrieval for structured signals has gained significant attention in recent years. A review of all of the results is beyond the scope of this paper, and we instead briefly mention some of the most relevant literature for the Gaussian measurement model. Oymak et. al. [30] analyzed the performance of the regularized PhaseLift algorithm and observed that the required sample complexity is of a suboptimal order compared to the optimal number of measurements required when phase information is available. For the special case of sparse phase retrieval similar results have been reported in [24] which indicates $\mathcal{O}(k^2 \log(n))$ measurements are required for recovering of a $k$-sparse signal, using regularized PhaseLift. Recently, there has been a stream of work on solving phase retrieval using non-convex methods [6, 47]. In particular, Soltanolkotabi [37] has shown that amplitude-based Wirtinger flow can break the $\mathcal{O}(k^2 \log(n))$ barrier. We also note that the paper [20] analyzed the PhaseMax algorithm with $\ell_1$ regularizer and observed that it achieves perfect recovery with $\mathcal{O}(k \log(n/k))$ samples, provided a well-correlated initialization point.

## 2 Preliminaries

### 2.1 Problem setup

Let $\mathbf{x}_0 \in \mathbf{R}^n$ denote the the underlying *structured* signal. We consider the *real* phase retrieval problem with the goal of recovering $\mathbf{x}_0$ from $m$ magnitude-only measurements of the form,

$$y_i = |\mathbf{a}_i^\mathsf{T} \mathbf{x}_0|, \ i = 1, 2, \ldots, m \,, \tag{1}$$

where $\{\mathbf{a}_i \in \mathbb{R}^n\}_{i=1}^m$ is the set of (known) measurement vectors. In practice, this set is identified based on the experimental settings; however, throughout this paper (for our analysis purposes) we assume that the $\mathbf{a}_i$'s are drawn independently from a Gaussian distribution with mean zero and covariance matrix $\mathbf{I}_n$. In order to exploit the structure of the signal we assume $f(\cdot)$ is a *convex* function that measures the "complexity" of the structured solution. The regularized PhaseMax algorithm also relies on an initial estimate of the true signal. Here, $\mathbf{x}_{\text{init}}$ is used to represent this initial guess. Our analysis is based on the critical assumption that both $\mathbf{x}_{\text{init}}$ and $\mathbf{x}_0$ are **independent** of all the measurement vectors. The constraint set in generalized PhaseMax is derived by simply relaxing the equality constraints in (1) into *convex* inequality constraints. We introduce the following convex optimization problem to recover the signal:

$$\hat{\mathbf{x}} = \underset{\mathbf{x} \in \mathbb{R}^n}{\operatorname{argmin}} \quad L_\lambda(\mathbf{x}) = -\mathbf{x}_{\text{init}}{}^\mathsf{T}\mathbf{x} + \lambda f(\mathbf{x})$$
$$\text{subject to:} \quad |\mathbf{a}_i^\mathsf{T}\mathbf{x}| \le y_i \,, \ \text{for } 1 \le i \le m. \tag{2}$$

The function $f$ is assumed to be sign invariant, i.e., $f(\mathbf{x}) = f(-\mathbf{x})$ for all $\mathbf{x} \in \mathbb{R}^n$ ($-\mathbf{x}$ has the same "complexity" as $\mathbf{x}$.) Note that because of the global phase ambiguity of measurements in (1), we can only estimate $\mathbf{x}_0$ up to a sign. Up to this sign ambiguity, we can use the normalized mean squared error (NMSE), defined as $\frac{||\hat{\mathbf{x}}-\mathbf{x}_0||^2}{||\mathbf{x}_0||^2}$, to measure the performance of the solution. In this paper we investigate the conditions under which the optimization program (2) uniquely identifies the true signal, i.e., $\hat{\mathbf{x}} = \mathbf{x}_0$ (up to the sign). Our results are asymptotic which is valid when $m, n \to \infty$.

### 2.2 Background on convex analysis

Our results give the required number of measurements as a function of certain geometrical properties of the descent cone of the objective function. Here, we recall these definitions from convex analysis.

**Definition 1.** *(Descent cone) For a function $R : \mathbb{R}^n \to \mathbb{R}$ the descent(tangent) cone at point $\mathbf{x}$ is defined as,*

$$T_R(\mathbf{x}) = cone(\{\mathbf{z} \in \mathbb{R}^n : R(\mathbf{x} + \mathbf{z}) \leq R(\mathbf{x})\}) , \tag{3}$$

*where $cone(\mathcal{S})$ denotes the closed conical hull of the set $\mathcal{S}$.*

**Definition 2.** *Let $\mathcal{S}$ be a closed convex set in $\mathbb{R}^n$. For $\mathbf{x} \in \mathbb{R}^n$ the projection of $\mathbf{x}$ on $\mathcal{S}$, denoted by $\Pi_{\mathcal{S}}(\mathbf{x})$, is defined as follows,*

$$\Pi_{\mathcal{S}}(\mathbf{x}) := \underset{\mathbf{y} \in \mathcal{S}}{argmin}||\mathbf{x} - \mathbf{y}|| , \tag{4}$$

*where $|| \cdot ||$ is the Euclidean norm. The distance function is defined as: $dist_{\mathcal{S}}(\mathbf{x}) = ||\mathbf{x} - \Pi_{\mathcal{S}}(\mathbf{x})||$.*

**Definition 3.** *(Statistical dimension) [1] The statistical dimension of a closed convex cone $\mathcal{C}$ in $\mathbb{R}^n$ is defined as,*

$$d(\mathcal{C}) = \mathbb{E}_{\mathbf{g}} \left[ ||\Pi_{\mathcal{C}}(\mathbf{g})||^2 \right] , \tag{5}$$

*where $\mathbf{g} \in \mathbb{R}^n$ is a standard normal vector.*

The statistical dimension canonically extends the dimension of linear spaces to convex cones. This quantity has been extensively studied in linear inverse problems. It is well-known that as $n \to \infty$, $m > d(T_{L_\lambda}(\mathbf{x}_0))$ is the necessary and sufficient condition for perfect signal recovery under noiseless linear Gaussian measurements [11, 38]. Our analysis indicates that given *phaseless* linear measurements, the regularized PhaseMax algorithm requires $\mathcal{O}(d(T_{L_\lambda}(\mathbf{x}_0)))$ measurements for perfect signal reconstruction. Therefore, it is order-wise optimal in that sense.

## 3 Main Results

In this section we present the main results of the paper which provide us with the required number of measurements for perfect signal recovery in the regularized PhaseMax optimization (2). This gives the value $m_0 = m_0(n, \mathbf{x}_0, \mathbf{x}_{\text{init}}, \lambda)$, such that the regularized PhaseMax algorithm uniquely identifies the underlying signal $\mathbf{x}_0$ with high probability whenever $m > m_0$.

In Section 3.1, we find sufficient conditions for recovery of the underlying signal. Theorem 1 provides an upper bound on the number of measurements that is equal to a constant factor times the statistical dimension of the descent cone, $d(T_{L_\lambda}(\mathbf{x}_0))$. Therefore, even though our analysis is not exact in this section, it leads us to the important observation that our proposed method is order-wise optimal in terms of the required sample complexity for perfect signal reconstruction.

In Section 3.2, we provide an exact analysis for the phase transition behavior of regularized PhaseMax when the regularizer is an absolutely scalable function. We apply this result to the case of unstructured phaseless recovery as well as sparse phaseless recovery to compute the exact phase transitions. We then compare the result of theory with the empirical results from numerical simulations.

### 3.1 Sufficient recovery condition

Let $\mathbf{P} := \frac{1}{||\mathbf{x}_0||^2} \mathbf{x}_0 \mathbf{x}_0^{\mathsf{T}}$ and $\mathbf{P}^{\perp} := \mathbf{I} - \mathbf{P}$ denote the projectors onto the span of $\mathbf{x}_0$ and its orthogonal complement, respectively, where $|| \cdot ||$ denotes the $\ell_2$-norm of the vectors. We also define $d^{(n)} := d(T_{L_\lambda}(\mathbf{x}_0))$ as the statistical dimension of the descent cone of the objective function at point $\mathbf{x}_0$. Our analysis rigorously characterizes the phase transition behavior of the regularized PhaseMax in the large system limit, i.e., when $n \to \infty$, while $m$ and $d^{(n)}$ grow at a proportional ratio $\delta = \frac{m}{d^{(n)}}$. $\delta$ is often called the oversampling ratio. Here, the superscript $(n)$ is used to denote the elements of a sequence. To streamline the notations, we often drop this when understood from the context.

Theorem 1 provides sufficient conditions for the successful recovery of $\mathbf{x}_0$. The recovery threshold depends on $\lambda$ and the initialization vector, $\mathbf{x}_{\text{init}}$. We define $\rho_{\text{init}} := \mathbf{x}_{\text{init}}^{\mathsf{T}} \mathbf{x}_0$ to quantify the caliber of the initial estimate. Due to the sign invariance property of the solution, we can assume without loss of generality that $\rho_{\text{init}} \geq 0$. Before stating the theorem, we shall introduce the function $R(\cdot) : (2, +\infty) \to \mathbb{R}_+$.

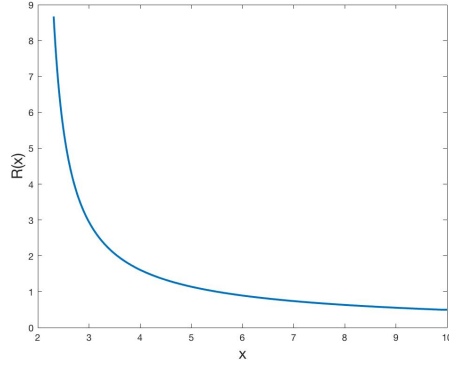

Figure 1: $R(x)$ for different values of $x$. $R$ is a monotonically decreasing function.

**Definition 4.** *For $x > 2$, $R(x)$ is the unique nonzero solution of the following equation:*

$$t^2 = \frac{x}{\pi}((1 + t^2)atan(t) - t) \; . \tag{6}$$

Figure 1 depicts the evaluation of the function $R(x)$ for different input values $x$. As observed, $R(x)$ is a decreasing function with respect to $x$, and it approaches zero as $x$ grows to infinity. It can be shown that for large values of the input $x$, $R(x)$ decays with the rate $\frac{1}{x}$.

**Theorem 1** (Sufficient recovery condition)**.** *For a fixed oversampling ratio $\delta > 2$, the regularized PhaseMax optimization* (2) *perfectly recovers the target signal (in the sense that $\lim_{n \to \infty} \mathbb{P}\{||\hat{\mathbf{x}} - \mathbf{x}_0||^2 > \epsilon||\mathbf{x}_0||^2\} = 0$, for any fixed $\epsilon > 0$) if,*

$$R(\delta) < \sup_{\mathbf{v} \in \partial L_\lambda(\mathbf{x}_0)} \frac{||\mathbf{P}\mathbf{v}||}{||\mathbf{P}^\perp \mathbf{v}||} \; , \tag{7}$$

*where $\partial L_\lambda(\mathbf{x}_0)$ denotes the sub-differential set of the objective function $L_\lambda(\cdot)$ at point $\mathbf{x}_0$.*

It is worth noting that $\partial L_\lambda(\mathbf{x}_0)$ is a convex and compact set, and it can be expressed in terms of the sub-differential of the regularization function $\partial f(\mathbf{x}_0)$ as following,

$$\partial L_\lambda(\mathbf{x}_0) = \{\lambda\mathbf{u} - \mathbf{x}_{\text{init}} : \mathbf{u} \in \partial f(\mathbf{x}_0)\} \; . \tag{8}$$

Observe that since $R(\cdot)$ is a monotonically decreasing function, the inequality (7) gives a lower bound for the oversampling ratio $\delta$. Indeed, we can restate the result in terms of this lower bound as the following corollary:

**Corollary 1.** *If there exist a fixed constant $\tau > 0$ such that,*

$$\sup_{\mathbf{v} \in \partial L_\lambda(\mathbf{x}_0)} \frac{||\mathbf{P}\mathbf{v}||}{||\mathbf{P}^\perp \mathbf{v}||} > \tau, \tag{9}$$

*then the regularized PhaseMax optimization* (2) *has perfect recovery for $\delta > C$, where $C$ is a constant that only depends on $\tau$.*

*Proof.* It is an immediate consequence of Theorem 1 by choosing $C = R^{-1}(\tau)$ and noting that $R(\cdot)$ is monotonically decreasing. $\qquad\square$

This result indicates that if $\mathbf{x}_{\text{init}}$ and $\lambda$ are chosen in such a way that the inequality (9) is satisfied for some positive constant $\tau$, then one needs $m > Cd^{(n)}$ measurement samples for perfect recovery, where $C$ is a constant and $d^{(n)}(= d)$ is the statistical dimension of the descent cone of the objective function at point $\mathbf{x}_0$. As motivating examples, we use Theorem 1 to find upper bounds on the phase transition when $\mathbf{x}_0$ has no structure or it is a sparse signal.

**Example 1:** Assume the target signal $\mathbf{x}_0$ has no a priori structure. The objective function in this case would be $L(\mathbf{x}) = -\mathbf{x}_{\text{init}}{}^\mathsf{T}\mathbf{x}$, and $\partial L(\mathbf{x}_0) = \{-\mathbf{x}_{\text{init}}\}$. It can be shown that the statistical dimension is

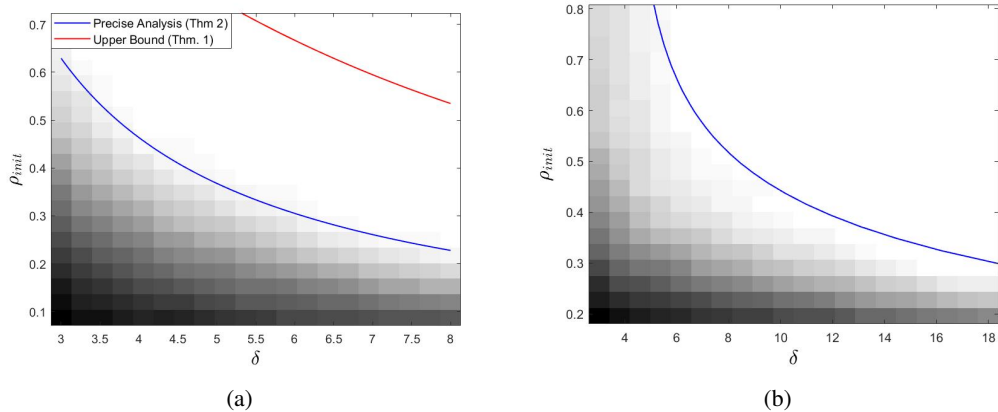

(a)                                                          (b)

Figure 2: Phase transition regimes for the regularized PhaseMax problem in terms of the oversampling ratio $\delta$ and $\rho_{\text{init}} = \mathbf{x}_{\text{init}}^{\mathsf{T}} \mathbf{x}_0$, for the cases of $\mathbf{x}_0$ with (a) no structure and (b) sparse signal recovery . The blue lines indicate the theoretical estimate for the phase transition derived from Theorem 2. The red line in (a) correspond to the upper bound calculated by Theorem 1. In the simulations we used signals of size $n = 128$. The result is averaged over 10 independent realization of the measurements.

$d^{(n)} = n - 1/2$. Due to the absence of the regularization term in this case, without loss of generality we can assume $\|\mathbf{x}_0\| = \|\mathbf{x}_{\text{init}}\| = 1$. Theorem 1 provides the following sufficient condition for perfect recovery:

$$\frac{||\mathbf{P}\mathbf{x}_{\text{init}}||}{||\mathbf{P}^{\perp}\mathbf{x}_{\text{init}}||} = \frac{\rho_{\text{init}}}{\sqrt{1 - \rho_{\text{init}}^2}} > R(\delta) \ . \tag{10}$$

This indicates $\mathcal{O}(n)$ measurements is sufficient for perfect recovery as long as $\rho_{\text{init}} \geq \rho_0$, where $\rho_0 > 0$ is a constant that does not approach zero as $n \to \infty$. The exact phase transition for the unstructured case (PhaseMax) has been derived in [14] which is compatible with this result. Figure 2(a) shows the result of numerical simulation for different values of $\delta$ and $\rho_{\text{init}}$, when $n = 128$. As depicted in the figure, the sufficient recovery condition from Theorem 1 is approximately a factor of 2 away from the actual phase transition.

**Example 2:** Let $\mathbf{x}_0$ be a $k$-sparse signal. In this case we use $|| \cdot ||_1$ as the regularization function. We show in Section 5.5 that if $\lambda > \frac{c}{\sqrt{k}}$, then $d^{(n)} \leq Ck \log(n/k)$, for some constants $c, C > 0$. This matches the well-known order for the statistical dimension derived in the compressive sensing literature [38].

Moreover, in order to satisfy the condition in Corollary 1 we need to have $\frac{\rho_{\text{init}}}{||\mathbf{x}_0||_1} > (1 + \epsilon)\lambda$, for some $\epsilon > 0$. Therefore, $\mathbf{x}_0$ can be perfectly recovered having $\mathcal{O}(k \log(n/k))$ samples when the hyper-parameter $\lambda$ is tuned properly, i.e., $\frac{c}{\sqrt{k}} < \lambda < \frac{\rho_{\text{init}}}{||\mathbf{x}_0||_1}$. Figure 3(a) compares this upper bounds with the precise analysis that we will show in Section 3.2. As depicted in this figure, the sufficient recovery condition is a valid upper bound on the phase transition, but it is not sharp.

### 3.2 Precise phase transition

So far, we have provided a sufficient condition for perfect signal recovery in the regularized PhaseMax. In this section we give the exact phase transition, i.e., the minimum number of measurements $m_0$ required for perfect recovery of the unknown vector $\mathbf{x}_0$. For our analysis, we assume that the function $f(\mathbf{x})$ is absolutely homogeneous (scalable), i.e., $f(\tau \cdot \mathbf{x}) = |\tau| \cdot f(\mathbf{x})$, for any scalar $\tau$. This covers a large range of regularization functions such as norms and semi-norms. Let $\partial L_\lambda^{\perp}(\mathbf{x}_0) \subset \mathbb{R}^n$ denote the projection of the sub-differential set into the orthogonal complement of $\mathbf{x}_0$, i.e.,

$$\partial L_\lambda^{\perp}(\mathbf{x}_0) = \{\mathbf{P}^{\perp}\mathbf{u} : \mathbf{u} \in \partial L_\lambda(\mathbf{x}_0)\} \ , \tag{11}$$

which is a convex and compact set. To state the result in a general framework, we require a further assumption on functions $L_\lambda^{(n)}(\cdot)$.

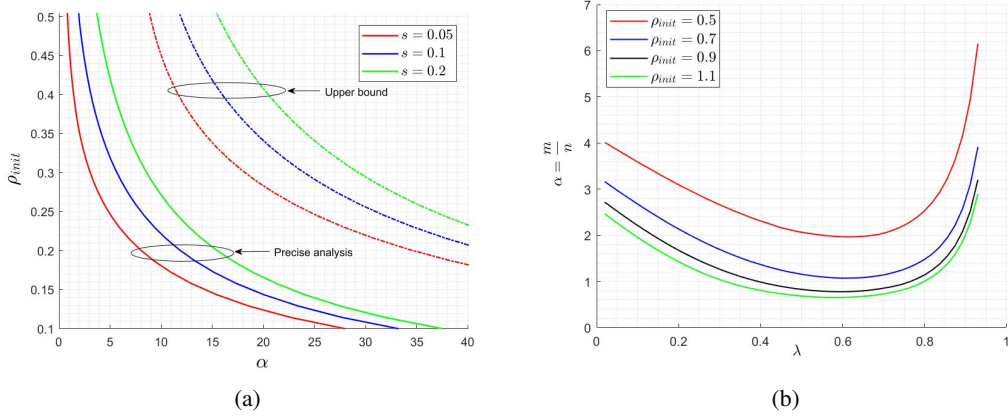

(a)                                                    (b)

Figure 3: (a) Comparing the upper bounds on the phase transition, derived by Theorem 1 (dashed lines) and the precise phase transition by Theorem 2 (solid lines), for three values of the sparsity factor $s = k/n$. (b) The phase transition behavior as a function of the regularization parameter $\lambda$, derived from the result of Theorem 2.

**Assumption 1** (Asymptotic functionals) *We say Assumption 1 holds if the following uniform convergences exist, as $n \to \infty$,*

$$\beta - \mathbb{E}\big[\frac{1}{\sqrt{n}} \mathbf{h}^{\mathsf{T}} \Pi_{\partial L_\lambda^\perp(\mathbf{x}_0)}(\frac{\beta}{\sqrt{n}}\mathbf{h})\big] \xrightarrow{Unif.} F_\lambda(\beta), \text{ and,}$$

$$\mathbb{E}\big[dist_{\partial L_\lambda^\perp(\mathbf{x}_0)}(\frac{\beta}{\sqrt{n}}\mathbf{h})\big] \xrightarrow{Unif.} G_\lambda(\beta) , \tag{12}$$

*where $\mathbf{h} \in \mathbb{R}^n$ has i.i.d. standard normal entries and $F_\lambda$, $G_\lambda : \mathbb{R}_+ \to \mathbb{R}$ denote the functions that the sequences uniformly converge to.*

One can show that, under some mild conditions on the regularization function $f(\cdot)$, Assumption 1 holds and also $F_\lambda(\beta) = G_\lambda(\beta)G'_\lambda(\beta)$, where $G'_\lambda(\cdot)$ denotes the derivative of the function $G_\lambda(\cdot)$. This assumption especially holds for the class of separable regularizers, where $f(\mathbf{v}) = \sum_i \tilde{f}(v_i)$ (e.g. $\ell_1$ norm for the case of sparse phase-retrieval). Later in this section, we will see validity of this assumption for two examples discussed earlier in Section 3. Our precise phase transition results indicate the required number of measurements as the solution of a set of two nonlinear equations with two unknowns. We define a new parameter $\alpha := \frac{m}{n}$, where $\alpha_{\text{opt}} = \frac{m_0}{n}$ indicates the exact phase transition of the regularized PhaseMax optimization. The following theorem gives an implicit formula to derive $\alpha_{\text{opt}}$.

**Theorem 2** (Precise phase transition). *Let $\hat{\mathbf{x}}$ be the solution to the regularized PhaseMax optimization (2) with the objective function $L_\lambda(\mathbf{x}) = -\mathbf{x}_{init}^{\mathsf{T}}\mathbf{x} + \lambda f(\mathbf{x})$, where the convex function $f(\cdot)$ is absolutely homogeneous and Assumption 1 holds. The regularized PhaseMax optimization would perfectly recover the target signal $\mathbf{x}_0$ if and only if:*

1. *$\alpha > \alpha_{opt}$, where $\alpha_{opt}$ is the solution of the following system of non-linear equations with two unknowns, $\alpha$ and $\beta$,*

$$\begin{cases} -G_\lambda(\beta) \, L_\lambda(\mathbf{x}_0) = \tan(\frac{\pi}{\alpha\beta}F_\lambda(\beta)) \, (G_\lambda^2(\beta) - \beta F_\lambda(\beta)) , \\ \tan(\frac{\pi}{\alpha\beta}F_\lambda(\beta)) \, (G_\lambda(\beta) + \frac{\pi}{\alpha\beta} F_\lambda(\beta) \, L_\lambda(\mathbf{x}_0)) = \frac{\pi}{\alpha\beta}F_\lambda(\beta) \, G_\lambda(\beta) , \end{cases} \tag{13}$$

2. *and, $L_\lambda(\mathbf{x}_0) < L_\lambda(0) = 0$ .*

*where the functions $F_\lambda(\cdot)$ and $G_\lambda(\cdot)$ are defined in (12).*

A few remarks are in place for this theorem:

**[Solving equations (13)]** The system of nonlinear equations (13) only involves two scalars $\beta$ and $\alpha$, and the functions $F_\lambda(\beta)$ and $G_\lambda(\beta)$ are determined by the objective function $L_\lambda(\mathbf{x})$. For our numerical simulations in the examples of Section 3.2.1 and Section 3.2.2, we used a fixed-point

iteration method that can quickly find the solution given a proper initialization.

**[Tuning $\lambda$]** Theorem 2 requires the objective function to satisfy $L_\lambda(\mathbf{x}_0) = \lambda f(\mathbf{x}_0) - \rho_{\text{init}} < 0$. Therefore, it is necessary to choose $\lambda$ in such a way that $\lambda < \rho_{\text{init}}/f(\mathbf{x}_0)$. Some additional assumptions on the unknown vector $\mathbf{x}_0$ enables us to calculate the proper range for $\lambda$. For instance, if we consider a random ensemble for $\mathbf{x}_0$ where the non-zero entries of $\mathbf{x}_0$ are Gaussian (or other) random variables, $\mathbb{E}[f(\mathbf{x}_0)]$ gives a reasonable estimation on $f(\mathbf{x}_0)$ that can help us choosing $\lambda$ appropriately. We will see an example of such case in section 3.2.2. Figure 3(b) shows an example of how the phase transition of the regularized PhaseMax, or equivalently the required sample complexity, behaves as a function of the hyper-parameter $\lambda$.

In the next sections, we use the result of Theorem 2 to compute the exact phase transition for the case of unstructured signal as well as the sparse signal recovery. Since the regularizer $f(\mathbf{x})$ is absolutely scalable, for both examples, we assume that $\|\mathbf{x}_0\| = 1$.

### 3.2.1  Unstructured signal recovery

When there is no a priori information about the structure of the target signal, we use the following optimization (PhaseMax) for signal recovery:

$$\hat{\mathbf{x}} = \underset{\mathbf{x} \in \mathbb{R}^n}{\operatorname{argmin}} \quad L(\mathbf{x}) = -\mathbf{x}_{\text{init}}^\mathsf{T} \mathbf{x}$$
$$\text{subject to:} \quad |\mathbf{a}_i^\mathsf{T} \mathbf{x}| \leq y_i , \quad \text{for } 1 \leq i \leq m . \tag{14}$$

Due to the absence of the regularization term, without loss of generality we can assume $\|\mathbf{x}_{\text{init}}\| = 1$. Moreover, $L(\mathbf{x}_0) = -\rho_{\text{init}}$ which indicates that the second condition in Theorem 2 . To apply the result of our theorem, we first compute explicit formulas for the functions $F_\lambda(\beta)$, and $G_\lambda(\beta)$, as follows,

$$F_\lambda(\beta) = \beta , \quad G_\lambda(\beta) = \sqrt{\beta^2 + 1 - \rho_{\text{init}}^2} . \tag{15}$$

We can now form the system of nonlinear equations (13) as follows,

$$\begin{cases} \sqrt{\beta^2 + 1 - \rho_{\text{init}}^2} \, \frac{\rho_{\text{init}}}{1 - \rho_{\text{init}}^2} = \tan(\frac{\pi}{\alpha}) , \\ \tan(\frac{\pi}{\alpha}) \, (\sqrt{\beta^2 + 1 - \rho_{\text{init}}^2} - \frac{\pi \rho_{\text{init}}}{\alpha}) = \frac{\pi}{\alpha} \sqrt{\beta^2 + 1 - \rho_{\text{init}}^2} . \end{cases} \tag{16}$$

Finally, solving equations (16) yields the following necessary and sufficient condition for perfect recovery,

$$\frac{\pi}{\alpha \tan(\pi/\alpha)} > 1 - \rho_{\text{init}}^2 , \tag{17}$$

which also verifies the result of [14].

Figure 2(a) shows the result of numerical simulations of running the PhaseMax algorithm for different values of $\rho_{\text{init}}$ and $\delta$. The intensity level of the color of each square in Figure 2, represents the error of PhaseMax in recovering $\mathbf{x}_0$. As seen in the figure, although our theoretical results has been established for the asymptotic setting (when the problem dimensions approach infinity), the blue line, which is derived from (17), reasonably predicts the phase transition for $n = 128$. The sufficient conditions that is derived from Theorem 1 is also depicted by the red line in the same figure.

### 3.2.2  Sparse recovery

We consider the case where the target signal $\mathbf{x}_0$ is sparse with $k$ non-zero entries. The convex function $f(\mathbf{x}) = \frac{1}{\sqrt{n}} \|\mathbf{x}\|_1$, which is known to be a proper regularizer that enforces sparsity [41], is used in the regularized PhaseMax optimization to recover $\mathbf{x}_0$,

$$\hat{\mathbf{x}} = \underset{\mathbf{x} \in \mathbb{R}^n}{\operatorname{argmin}} \quad L_\lambda(\mathbf{x}) = -\mathbf{x}_{\text{init}}^\mathsf{T} \mathbf{x} + \frac{\lambda}{\sqrt{n}} \|\mathbf{x}\|_1$$
$$\text{subject to:} \quad |\mathbf{a}_i^\mathsf{T} \mathbf{x}| \leq y_i , \quad \text{for } 1 \leq i \leq m . \tag{18}$$

To streamline notations, we assume the non-zero entries of $\mathbf{x}_0$ are the first $k$ entries and decompose vector $\mathbf{v} \in \mathbb{R}^n$ as $\mathbf{v} = \begin{bmatrix} \mathbf{v}^\Delta \\ \mathbf{v}^{\Delta^c} \end{bmatrix}$, where $\mathbf{v}^\Delta \in \mathbb{R}^k$ denotes the first $k$ entries of $\mathbf{v}$, and $\mathbf{v}^{\Delta^c} \in \mathbb{R}^{n-k}$ is the remaining $n - k$ entries. As $m, n \to \infty$, we would like to apply the result of Theorem 2

to compute the exact phase transition. Due to the rotational invariance property of the Gaussian distribution, it can be shown that multiplying the last $(n-k)$ entries of $\mathbf{x}_{\text{init}}$, by a unitary matrix $\mathbf{U} \in \mathbb{R}^{(n-k)\times(n-k)}$ does not change the phase transition behavior in (2). Hence, we can assume the entries of $\mathbf{x}_{\text{init}}^{\Delta^c}$ have Gaussian distribution, i.e.,

$$\mathbf{x}_{\text{init}} = \begin{bmatrix} \mathbf{x}_{\text{init}}^{\Delta} \\ \mathbf{x}_{\text{init}}^{\Delta^c} \end{bmatrix}, \quad \text{and} \quad \mathbf{x}_{\text{init}}^{\Delta^c} = \frac{1}{\sqrt{n-k}}\,\|\mathbf{x}_{\text{init}}^{\Delta^c}\|\,\mathbf{g}\,, \tag{19}$$

where $\mathbf{g} \in \mathbb{R}^{n-k}$ has standard normal entries. This observation enables us to establish the following lemma:

**Lemma 1.** *Consider the optimization problem* (18) *to recover the $k$-sparse signal $\mathbf{x}_0$. We assume the entries of $\mathbf{x}_{init}$ are distributed as in* (19) *and define $\tilde{\rho} := \frac{1}{\sqrt{k}}sign(\mathbf{x}_0^{\Delta})^{\mathsf{T}}\mathbf{x}_{init}^{\Delta}$, where $sign(\cdot)$ denotes the component-wise sign function. Then, Assumption 1 holds with:*

$$F_\lambda(\beta) = \beta\left(s + 2(1-s)\cdot Q\left(\frac{\lambda}{\sqrt{\beta^2 + \frac{\|\mathbf{x}_{init}^{\Delta^c}\|^2}{1-s}}}\right)\right),$$

$$G_\lambda^2(\beta) = s\cdot(\beta^2 + \lambda^2) + \|\mathbf{x}_{init}^{\Delta}\|^2 - 2\lambda\sqrt{s}\tilde{\rho} - L^2(\mathbf{x}_0)$$

$$+ (1-s)\left(\beta^2 + \frac{\|\mathbf{x}_{init}^{\Delta^c}\|^2}{1-s}\right)\cdot\mathbb{E}_H\left[\,shrink^2\left(H, \frac{\lambda}{\sqrt{\beta^2 + \frac{\|\mathbf{x}_{init}^{\Delta^c}\|^2}{1-s}}}\right)\,\right] \tag{20}$$

*where $Q(\cdot)$ is the tail distribution of the standard normal distribution, $H$ has standard normal distribution and $s := k/n$ is the sparsity factor. The shrinkage function $shrink(\cdot,\cdot) : \mathbb{R} \times \mathbb{R}_+ \to \mathbb{R}_+$ is defined as:*

$$shrink(x,\tau) = (|x| - \tau)\mathbf{1}\{|x| \geq \tau\}\,. \tag{21}$$

It is worth noting that the function $shrink(\cdot,\cdot)$ also appeared in computing the statistical dimension for $\ell_1$ regularization (see Section 5.5) which indicates some implicit relation to $\alpha_{opt}$.

We have numerically computed the solution of the nonlinear system (20). Figure 2(a), and Figure 2(b) shows the error of regularized PhaseMax over a range of $\rho_{\text{init}}$ and $\delta$. The comparison between our upper bound derived from Theorem 1 and precise analysis of Theorem 2 is depicted in Figure 3(a) for three values of the sparsity factor $s = 0.05, 0.1, 0.2$. Observe that the upper bound is only a constant factor away from the precise phase transition, while its derivation involves simpler formulas. Finally, Figure 3(b), illustrates impact of the regularization parameter $\lambda$ on the phase transition of the regularized PhaseMax optimization for four values of $\rho_{\text{init}}$. The values of $\lambda$ in this figure are normalized by $\frac{\rho_{\text{init}}\sqrt{n}}{\|\mathbf{x}_0\|}$, which is the maximum acceptable value of $\lambda$ in the regularized PhaseMax.

## 4 Conclusion and Future Directions

In this paper, we introduced a new convex optimization framework, *regularized PhaseMax*, to solve the structured phase retrieval problem. We have shown that, given a proper initialization, the regularized PhaseMax optimization perfectly recovers the underlying signal from a number of phaseless measurements that is only a constant factor away from the number of measurements required when the phase information is available. We explicitly computed this constant factor.

An important (yet still open) research problem is to investigate the required sample complexity to construct a proper initialization vector, $\mathbf{x}_{\text{init}}$. As an example, for the case of sparse phase retrieval, even though our analysis indicates that $\mathcal{O}(k\log\frac{n}{k})$ is the required sample complexity of the regularized PhaseMax optimization, the best known initialization technique [6] needs $\mathcal{O}(k^2\log n)$ samples to generate a meaningful initialization, which is suboptimal. An important future direction is to study initialization techniques that break this sample complexity barrier, or to use information theoretic arguments (as in [28]) to show that the sample complexity for the initialization cannot be improved.

To form the objective function in the regularized PhaseMax, we exploited some a-priori knowledge about the structure of the underlying signal. In many practical settings such prior information is not available. There has been some interesting recent publications (e.g. [4, 48]) which introduce efficient algorithms to learn the structure of the underlying signal. An interesting research direction is to investigate new optimization framework that does not rely on the prior information about the structure of the underlying signal.

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
