[Supplementary Material]

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

$$\Phi(\mathbf{G}) := \min_{\mathbf{w} \in \mathbf{S_w}} \max_{\mathbf{u} \in \mathbf{S_u}} \mathbf{u}^\mathsf{T} \mathbf{G} \mathbf{w} + \psi(u, w), \tag{22a}$$

$$\phi(\mathbf{g}, \mathbf{h}) := \min_{\mathbf{w} \in \mathbf{S_w}} \max_{\mathbf{u} \in \mathbf{S_u}} \|\mathbf{w}\| \mathbf{g}^\mathsf{T} \mathbf{u} - \|\mathbf{u}\| \mathbf{h}^\mathsf{T} \mathbf{w} + \psi(u, w), \tag{22b}$$

where $\mathbf{G} \in \mathbb{R}^{m \times n}, \mathbf{g} \in \mathbb{R}^m, \mathbf{h} \in \mathbb{R}^n, \mathbf{S_w} \subset \mathbb{R}^n, \mathbf{S_u} \subset \mathbb{R}^m$ and $\psi : \mathbb{R}^n \times \mathbb{R}^m \to \mathbb{R}$. Denote $\mathbf{w}_\Phi := \mathbf{w}_\Phi(\mathbf{G})$ and $\mathbf{w}_\phi := \mathbf{w}_\phi(\mathbf{g}, \mathbf{h})$ any optimal minimizers in (22a) and (22b), respectively. The following lemma is a result of CGMT [39].

**Lemma 2.** *Consider the two optimizations* (22a) *and* (22b)*. Let* $\mathbf{S_w}, \mathbf{S_u}$ *be convex sets where at least one of them is compact,* $\psi$ *be continuous and convex-concave on* $\mathbf{S_w} \times \mathbf{S_u}$*, and,* $\mathbf{G}, \mathbf{g}$ *and* $\mathbf{h}$ *all have entries iid standard normal. Suppose there exist a unique* $\alpha$ *such that in the limit of* $n \to \infty$ *it holds in probability that* $\|\mathbf{w}_\phi(\mathbf{g}, \mathbf{h})\| \to \alpha$*. Then, the same holds for* $\mathbf{w}_\Phi(\mathbf{G})$ *and we have* $\|\mathbf{w}_\Phi(\mathbf{G})\| \to \alpha$*.*

CGMT essentially replaces the optimization in (22a) with the one in (22b) (which we refer to as the Auxiliary Optimization) which is simpler to analyze. This is the main step in our proofs.

### 5.1.1 Convergence analysis of the Auxiliary Optimization

After applying CGMT in the proof of Theorem 2, we aim to analyze the auxiliary optimization. In this path, we replace several functions with their limits in probability. This can be done through the same tricks used in section A.4 of [39] and Lemma B.1 in the same paper. Here, we state the following lemma without proof.

**Lemma 3** (Min-convergence – Open Sets). *Consider a sequence of proper, convex stochastic functions* $M_n : (0, \infty) \to \mathcal{R}$*, and, a deterministic function* $M : (0, \infty) \to \mathcal{R}$*, such that:*

1. *$M_n(x) \xrightarrow{P} M(x)$, for all $x > 0$,*

2. *there exists $z > 0$ such that $M(x) > \inf_{x>0} M(x)$ for all $x \geq z$.*

*Then,* $\inf_{x>0} M_n(x) \xrightarrow{P} \inf_{x>0} M(x)$*.*

The objective function in our optimization problems satisfy the assumptions of this lemma at the points that we replace them with their limits.

## 5.2 Proof of Theorem 1

In order to establish the result we use the following lemma which provides an equivalent optimization that has the same error performance as PhaseMax, and is the key ingredient in deriving the main results of the paper.

**Lemma 4** (Equivalent Optimization). *Consider the regularized PhaseMax problem introduced in Section 2.1. As* $n \to \infty$*, the error performance converges in probability as follows:*

$$\left( \frac{\|\hat{\mathbf{x}} - \mathbf{x}_0\|^2}{\|\mathbf{x}_0\|^2} \right) - \left( \|\mathbf{w}^\star\|^2 + (1 - s^\star)^2 \right) \xrightarrow{n \to \infty} 0 . \tag{23}$$

*Here* $s^\star \in \mathbb{R}$*, and* $\mathbf{w}^\star \in \mathbb{R}^n$ *are the unique optimizers of the following optimization program,*

$$\min_{s \in \mathbb{R}} \min_{\mathbf{w} \in \mathbb{R}^n, \mathbf{w} \perp \mathbf{x}_0} \quad -\mathbf{x}_{init}{}^\mathsf{T}(s\mathbf{x}_0 + \mathbf{w}) + \lambda f(s\mathbf{x}_0 + \mathbf{w})$$
$$\text{subject to:} \quad \mathbf{h}^\mathsf{T}\mathbf{w} \geq \sqrt{m\, \mathbf{c_d}(s, \|\mathbf{w}\|)} \quad , \tag{24}$$

*where $\mathbf{h} \in \mathbb{R}^n$ has i.i.d. standard normal entries and the function $\mathbf{c_d} : \mathbb{R} \times \mathbb{R}_+ \to \mathbb{R}$ is defined as,*

$$\mathbf{c_d}(s, r) = \frac{1}{\pi}[((1+s)^2 + r^2) \; atan(\frac{r}{1+s}) \; + ((1-s)^2 + r^2) \; atan(\frac{r}{1-s}) - 2r]. \qquad (25)$$

The full technical details of obtaining this result is explained in Section 5.3. In short, to show the equivalence, we start from (2) and define new variables $s := \mathbf{x}_0^\mathsf{T}\mathbf{x}$ and $\mathbf{w} = \mathbf{P}^\perp\mathbf{x}$. Then reformulate it as an unconstrained optimization using Lagrange multipliers. The result is a consequence of applying CGMT (Lemma 2, see Appendix 5.1) with some simplifications.

Consider the following optimization:

$$\min_{s \in \mathbb{R}} \min_{\mathbf{w} \in \mathbb{R}^n, \mathbf{w} \perp \mathbf{x}_0} \quad -\mathbf{x}_{\text{init}}^\mathsf{T}(s\mathbf{x}_0 + \mathbf{w}) + \lambda f(s\mathbf{x}_0 + \mathbf{w})$$
$$\text{subject to:} \quad \mathbf{h}^\mathsf{T}\mathbf{w} \geq \sqrt{m \; \mathbf{c_d}(s, ||\mathbf{w}||)} \; . \qquad (26)$$

The result of Lemma 4 established that as $n \to \infty$, the error performance of the regularized PhaseMax converges to the error performance in (26). The following corollary indicates the necessary and sufficient condition for perfect recovery:

**Corollary 2.** *As $n \to \infty$, $\mathbf{x}_0$ is the unique solution of the regularized PhaseMax optimization, if and only if $(s^\star, \mathbf{w}^\star) = (1, \mathbf{0})$ be the unique optimizer of the equivalent optimization (26).*

*Proof.* This is an immediate result of Lemma 4, noticing that $||\hat{\mathbf{x}} - \mathbf{x}_0|| = 0$ is the condition for perfect recovery. $\qquad \square$

Let us proceed onwards with analyzing (26). For simplicity, we assume $||\mathbf{x}_0|| = 1$. Define a new function $\hat{f}(\cdot) : \mathbb{R}^n \to \mathbb{R}$ as follows,

$$\hat{f}(\mathbf{x}) = f(\mathbf{x}_0) + \max_{\mathbf{v} \in \partial f(\mathbf{x}_0)} \mathbf{v}^\mathsf{T}(\mathbf{x} - \mathbf{x}_0) \; . \qquad (27)$$

$\partial f(\mathbf{x}_0)$ is the sub-differential set of function $f$ at point $\mathbf{x}_0$ which is a convex and compact set. $\hat{f}(\cdot)$ is basicly the first-order approximation of the regularization function $f(\cdot)$ at point $\mathbf{x}_0$. This replacement cannot be done in general, but since we are only investigating the phase transition regime where the norm of the error, $||\hat{\mathbf{x}} - \mathbf{x}_0||$, approaches to zero, we may perform this exchange. To investigate the phase transition behavior in (26), we bound $1 - s$ and $||\mathbf{w}||$ to a small neighborhood of 0. Therefore, it is valid to replace $f$ with $\hat{f}$ in that small neighborhood around $\mathbf{x}_0$. Reformulating the optimization using this replacement would give us the following,

$$\min_{s \in \mathbb{R}, \mathbf{w} \perp \mathbf{x}_0} \max_{\mathbf{v} \in \partial f(\mathbf{x}_0)} \quad -\mathbf{x}_{\text{init}}^\mathsf{T}(s\mathbf{x}_0 + \mathbf{w}) + \lambda f(\mathbf{x}_0) + \lambda \mathbf{v}^\mathsf{T}((s-1)\mathbf{x}_0 + \mathbf{w})$$
$$\text{subject to:} \quad \mathbf{h}^\mathsf{T}\mathbf{w} \geq \sqrt{m \; \mathbf{c_d}(s, ||\mathbf{w}||)} \; . \qquad (28)$$

We add the constant term, $\mathbf{x}_{\text{init}}^\mathsf{T}\mathbf{x}_0 - \lambda f(\mathbf{x}_0)$, to the objective function and reformulate the maximization in terms of $\partial L(\mathbf{x}_0)$ as follows,

$$\min_{s \in \mathbb{R}, \mathbf{w} \perp \mathbf{x}_0} \max_{\mathbf{v} \in \partial L(\mathbf{x}_0)} \quad (s-1)\mathbf{x}_0^\mathsf{T}\mathbf{v} + \mathbf{w}^\mathsf{T}\mathbf{v}$$
$$\text{subject to:} \quad \mathbf{h}^\mathsf{T}\mathbf{w} \geq \sqrt{m \; \mathbf{c_d}(s, ||\mathbf{w}||)} \; . \qquad (29)$$

If $|s| > 1$ in (29), we have the following inequalities:

$$||\mathbf{w}||^2 \; d(T_L(\mathbf{x}_0)) \geq (\mathbf{h}^\mathsf{T}\mathbf{w})^2 \geq m \; \mathbf{c_d}(s, ||\mathbf{w}||) > \frac{m}{2}||\mathbf{w}||^2 \; . \qquad (30)$$

The first inequality is due to the fact that $\mathbf{x} - \mathbf{x}_0 = (s-1)\mathbf{x}_0 + \mathbf{w}$ is in $T_L(\mathbf{x}_0)$ (the descent cone of the objective at point $\mathbf{x}_0$). The second inequality appeared as a constraint in the optimization

problem (29). The last inequality is true since $\mathbf{c_d}(s, r) > r^2/2$, when $|s| \geq 1$. Therefore, using the assumption $\delta = \frac{m}{d} > 2$, it can be shown that the feasible set of (29) is nonempty if and only if $|s| \leq 1$.

Since the regularized PhaseMax optimization is convex, in order to show that $s^\star = 1$ and $\mathbf{w}^\star = \mathbf{0}$ are the unique optimizers of (29), it is sufficient to check the optimality condition in a small neighborhood of $(s^\star = 1, \mathbf{w}^\star = \mathbf{0})$. We also use the following approximation of the function $\mathbf{c_d}(s, r)$ which is valid in a small neighborhood around the point $(s, r) = (1, 0)$:

$$\mathbf{c_d}(s, r) = \frac{1}{\pi}[((1 - s)^2 + r^2)\mathrm{atan}(\frac{r}{1 - s}) - r(1 - s)] . \tag{31}$$

Next, for fixed $|s| < 1$, we will find an upper bound for $r := ||\mathbf{w}||$ such that $s$ and $\mathbf{w}$ satisfy the constraint in (29). We use the following inequalities:

$$r^2 \, d(T_L(\mathbf{x}_0)) \geq (\mathbf{h}^\mathsf{T}\mathbf{w})^2 \geq m \, \mathbf{c_d}(s, r) \Rightarrow r^2 \geq \delta \, \mathbf{c_d}(s, r) . \tag{32}$$

Replacing the approximation (31) for $\mathbf{c_d}(s, r)$, when $s \uparrow 1$ we have,

$$r \leq R(\delta)(1 - s) , \tag{33}$$

where $R(\delta)$ is the unique nonzero solution of the following nonlinear equation:

$$t^2 = \frac{\delta}{\pi}((1 + t^2)\mathrm{atan}(t) - t) . \tag{34}$$

We are now at the stage to establish the result of Theorem 1. Assume $\tilde{\mathbf{v}} \in \partial L(\mathbf{x}_0)$ achieves the supremum in (7) (Note that $\tilde{\mathbf{v}}$ always exist because the set $\partial L(\mathbf{x}_0)$ is compact). $\tilde{\mathbf{v}}$ then satisfies the following conditions:

1. $\mathbf{x}_0^\mathsf{T}\tilde{\mathbf{v}} < 0$ ,
2. $||\mathbf{P}\tilde{\mathbf{v}}|| > R(\delta) \, ||\mathbf{P}^\perp\tilde{\mathbf{v}}||$ .

We have the following inequalities:

$$
\begin{aligned}
\min_{|s| \leq 1, \mathbf{w} \perp \mathbf{x}_0} \max_{\mathbf{v} \in \partial L(\mathbf{x}_0)} (s - 1)\mathbf{x}_0^\mathsf{T}\mathbf{v} + \mathbf{w}^\mathsf{T}\mathbf{v} \quad &\geq \quad \min_{|s| \leq 1, \mathbf{w} \perp \mathbf{x}_0} (s - 1)\mathbf{x}_0^\mathsf{T}\tilde{\mathbf{v}} + \mathbf{w}^\mathsf{T}\tilde{\mathbf{v}} \\
&\geq \quad \min_{|s| \leq 1, \mathbf{w} \perp \mathbf{x}_0} (1 - s)||\mathbf{P}\tilde{\mathbf{v}}|| - ||\mathbf{w}|| \, ||\mathbf{P}^\perp\tilde{\mathbf{v}}|| ,
\end{aligned}
\tag{35}
$$

where for the first inequality, we used the fact that maximization over $\mathbf{v}$ gives a larger value compare to choosing the specific vector $\tilde{\mathbf{v}}$. The second inequality we used Cauchy-Schwarz to bound $\mathbf{w}^\mathsf{T}\mathbf{v}$ from below. When $s \uparrow 1$ we use the approximation (32) which bounds $||\mathbf{w}||$ from above. Therefore, we have:

$$(1 - s)||\mathbf{P}\tilde{\mathbf{v}}|| - ||\mathbf{w}|| \, ||\mathbf{P}^\perp\tilde{\mathbf{v}}|| > (1 - s)(||\mathbf{P}\tilde{\mathbf{v}}|| - R(\delta)||\mathbf{P}^\perp\tilde{\mathbf{v}}||) > 0. \tag{36}$$

And this would give the final result that $s^\star = 1$, $\mathbf{w}^\star = \mathbf{0}$ is the unique solution of (29). The perfect recovery in the generalized PhaseMax follows from the result of Corollary 2.

## 5.3 Proof of Lemma 4

Define matrix $\mathbf{A} \in \mathbb{R}^{m \times n}$ with $i^{\mathrm{th}}$ row equal to the measurement vector $\mathbf{a}_i$, for $i = 1, 2, \ldots, m$. Let $\mathbf{y} := |\mathbf{A}\mathbf{x}_0| \in \mathbb{R}^m$ denote the measurement values. To streamline our analysis, we assume $||\mathbf{x}_0|| = 1$. One can rewrite the constraint set of the optimization problem (2) as followings,

$$|\mathbf{A}\mathbf{x}| \leq \mathbf{y} \quad \Leftrightarrow \quad -\mathbf{A}\mathbf{x} + \mathbf{y} \geq \mathbf{0} , \text{ and } \mathbf{A}\mathbf{x} + \mathbf{y} \geq \mathbf{0}, \tag{37}$$

where all the inequalities are component-wise. Exploiting the Lagrange multipliers, we can reformulate the generalized PhaseMax optimization as,

$$\min_{\mathbf{x}\in\mathbb{R}^n} \max_{\boldsymbol{\mu},\boldsymbol{\eta}\in\mathbb{R}^m_+} \quad -\mathbf{x}_{\text{init}}^\mathsf{T}\mathbf{x} + \lambda f(\mathbf{x}) + (\boldsymbol{\mu}-\boldsymbol{\eta})^\mathsf{T}\mathbf{A}\mathbf{x} - (\boldsymbol{\mu}+\boldsymbol{\eta})^\mathsf{T}\mathbf{y} , \tag{38}$$

where $\mu_i$ and $\eta_i$ are Lagrange multipliers for the inequalities $\mathbf{a}_i^\mathsf{T}\mathbf{x} \leq y_i$ and $\mathbf{a}_i^\mathsf{T}\mathbf{x} \geq -y_i$, respectively. Assume $y_i > 0$ (which happens almost surely), these two inequalities cannot be active at the same time. Therefore, at least one of $\mu_i$ and $\eta_i$ must be equal to 0, for every $i = 1, 2, \ldots, m$. Hence, we have $\boldsymbol{\mu}+\boldsymbol{\eta} = |\boldsymbol{\mu}-\boldsymbol{\eta}|$. Here $|\cdot|$ denotes the component-wise absolute value function. Define $\mathbf{v} := \boldsymbol{\mu}-\boldsymbol{\eta} \in \mathbb{R}^m$ and rewrite the optimization in terms of $\mathbf{v}$ gives the following,

$$\min_{\mathbf{x}\in\mathbb{R}^n} \max_{\mathbf{v}\in\mathbb{R}^m} \quad -\mathbf{x}_{\text{init}}^\mathsf{T}\mathbf{x} + \lambda f(\mathbf{x}) + \mathbf{v}^\mathsf{T}\mathbf{A}\mathbf{x} - |\mathbf{v}|^\mathsf{T}|\mathbf{A}\mathbf{x}_0| . \tag{39}$$

Since the term $|\mathbf{v}|^\mathsf{T}|\mathbf{A}\mathbf{x}_0|$ depends on the matrix $\mathbf{A}$, it is not possible to apply the CGMT to the bilinear form $\mathbf{v}^\mathsf{T}\mathbf{A}\mathbf{x}$. In order to apply CGMT, we use the following key decomposition for $\mathbf{x}$:

$$\mathbf{x} = s\mathbf{x}_0 + \mathbf{w}, \tag{40}$$

where $s = \mathbf{x}_0^\mathsf{T}\mathbf{x} \in \mathbb{R}$ is a scalar and the vector $\mathbf{w} = \mathbf{P}^\perp\mathbf{x} \in \mathbb{R}^n$ is orthogonal to $\mathbf{x}_0$. We can rewrite the optimization problem (39) in terms of $s$ and $\mathbf{w}$,

$$\min_{s\in\mathbb{R},\ \mathbf{w}\perp\mathbf{x}_0} \max_{\mathbf{v}\in\mathbb{R}^m} \quad -s\rho_{\text{init}} - \mathbf{x}_{\text{init}}^\mathsf{T}\mathbf{w} + \lambda f(s\mathbf{x}_0 + \mathbf{w}) + \mathbf{v}^t\mathbf{A}\mathbf{w} + s\mathbf{v}^\mathsf{T}\mathbf{A}\mathbf{x}_0 - |\mathbf{v}|^\mathsf{T}|\mathbf{A}\mathbf{x}_0|, \tag{41}$$

where $\rho_{\text{init}} = \mathbf{x}_{\text{init}}^\mathsf{T}\mathbf{x}_0$. Next, we use the following property of Gaussian matrices.

**Lemma 5.** *Let $\mathbf{G} \in \mathbb{R}^{m\times n}$ be a random matrix with i.i.d. standard normal entries, and $\mathbf{u}, \mathbf{v} \in \mathbb{R}^n$ are such that $\mathbf{u} \perp \mathbf{v}$. The random vectors $\mathbf{G}\mathbf{u}$ and $\mathbf{G}\mathbf{v}$ are independent.*

*Proof.* Let $\mathbf{G} = [g_{i,j}]_{m\times n}$ and define $\mathbf{a} = \mathbf{G}\mathbf{u}$, and $\mathbf{b} = \mathbf{G}\mathbf{v}$. Since $\mathbf{G}$ has Gaussian entries $\mathbf{a}, \mathbf{b}$ are Gaussian vectors in $\mathbb{R}^m$. Therefore, to show their independence it is sufficient to show that $\mathbb{E}[\mathbf{a}\mathbf{b}^\mathsf{T}] = \mathbf{0}_{m\times m}$.

$$\mathbb{E}[a_i b_j] = \sum_{k=1}^n \sum_{l=1}^n u_k v_l \, \mathbb{E}[g_{i,k}\ g_{j,l}] = \begin{cases} \sum_{k=1}^n u_k v_k = 0, & \text{if } i = j \\ 0 , & \text{if } i \neq j \end{cases} , \tag{42}$$

where we used the fact that $\mathbf{u}^\mathsf{T}\mathbf{v} = \sum_{k=1}^n u_k v_k = 0$.

$\square$

Using the result of Lemma 5 , the random vectors $\mathbf{A}\mathbf{x}_0$ and $\mathbf{A}\mathbf{w}$ are independent. So, we are allowed to change the matrix $\mathbf{A}$ in the bilinear form $\mathbf{v}^\mathsf{T}\mathbf{A}\mathbf{w}$ with its independent copy $\mathbf{H} \in \mathbb{R}^{m\times n}$ which also has i.i.d. standard normal entries. We also define $\mathbf{q} = \mathbf{A}\mathbf{x}_0 \in \mathbb{R}^m$, which is independent of $\mathbf{H}$. Note that since $\mathbf{A}$ has i.i.d. normal entries and $\|\mathbf{x}_0\| = 1$, the entries of $\mathbf{q}$ also has i.i.d. standard normal distribution. We can rewrite the optimization as follows:

$$\min_{s\in\mathbb{R},\ \mathbf{w}\perp\mathbf{x}_0} \max_{\mathbf{v}\in\mathbb{R}^m} \quad -s\rho_{\text{init}} - \mathbf{x}_{\text{init}}^\mathsf{T}\mathbf{w} + \lambda f(s\mathbf{x}_0 + \mathbf{w}) + \mathbf{v}^t\mathbf{H}\mathbf{w} + s\mathbf{v}^\mathsf{T}\mathbf{q} - |\mathbf{v}|^\mathsf{T}|\mathbf{q}| . \tag{43}$$

Now, we would like to apply the CGMT framework in Lemma 2 to equation (43), in order to replace the bilinear form $\mathbf{v}^\mathsf{T}\mathbf{H}\mathbf{w}$ with two linear forms $\|\mathbf{v}\|\mathbf{h}^\mathsf{T}\mathbf{w} + \mathbf{v}^\mathsf{T}\mathbf{g}\|\mathbf{w}\|$. But this lemma requires the set that we optimize $\mathbf{w}$ over to be compact. In order to be able to apply CGMT, we enforce an "artificial" bound on the norm of $\mathbf{w}$. Note that our goal is to prove that eventually, $\hat{\mathbf{w}}$ converges to a finite number $\alpha^\star$. We define $K_\alpha = \alpha^\star + \Delta$ for some $\Delta > 0$ and also the compact set $\mathcal{S}_\mathbf{w} = \{\mathbf{w}|\mathbf{w} \perp \mathbf{x}_0 , \|\mathbf{w}\| \leq K_\alpha\}$. Let $\hat{\mathbf{w}}^{\text{temp}}$ to be the optimizer to the version of (43) where we

optimize $\mathbf{w}$ over $\mathcal{S}_{\mathbf{w}}$. It is simple to verify that if $\|\hat{\mathbf{w}}^{\text{temp}}\| \xrightarrow{\text{P}} \alpha^{\star}$, then $\|\hat{\mathbf{w}}\| \xrightarrow{\text{P}} \alpha^{\star}$. This means that if in the final equation, we get a unique finite solution for the asymptotic behaviour of $\|\hat{\mathbf{w}}\|$ (which is what we do) , the proof goes though and we can apply CGMT.

Now that this concern is taken care of, the following corollary will be the result of applying CGMT to the equation (43).

**Corollary 3.** *Let $\hat{\mathbf{x}}$ be the unique optimizer of the generalized PhaseMax algorithm* (2)*. As $n \to \infty$ the error performance converges in probability as follows:*

$$\left( \frac{\|\hat{\mathbf{x}} - \mathbf{x}_0\|^2}{\|\mathbf{x}_0\|^2} \right) - \left( \|\mathbf{w}^{\star}\|^2 + (1 - s^{\star})^2 \right) \xrightarrow{n \to \infty} 0 , \tag{44}$$

*where $s^{\star}$, $\mathbf{w}^{\star}$ are the unique optimizers of the following (auxiliary) optimization:*

$$\min_{s \in \mathbb{R}, \ \mathbf{w} \perp \mathbf{x}_0} \ \max_{\mathbf{v} \in \mathbb{R}^m} \ - s\rho_{init} - \mathbf{x}_{init}^{\mathsf{T}}\mathbf{w} + \lambda f(s\mathbf{x}_0 + \mathbf{w}) - \|\mathbf{v}\| \|\mathbf{h}^{\mathsf{T}}\mathbf{w} + \mathbf{v}^{\mathsf{T}}\mathbf{g}\| \|\mathbf{w}\| + s\mathbf{v}^{\mathsf{T}}\mathbf{q} - \|\mathbf{v}\|^{\mathsf{T}}\|\mathbf{q}\| . \tag{45}$$

$\mathbf{h} \in \mathbb{R}^n$ *and* $\mathbf{g} \in \mathbb{R}^m$ *are random vectors with i.i.d. standard normal entries.*

We proceed onwards with analyzing (45). Observe that if we fix $|\mathbf{v}|$, then the optimal $\mathbf{v}$ satisfies $\text{sign}(\mathbf{v}) = \text{sign}(\|\mathbf{w}\|\mathbf{g} + s\mathbf{q})$ ($\text{sign}(\cdot)$ are component-wise functions) which simplifies the optimization to the following,

$$\min_{s \in \mathbb{R}, \ \mathbf{w} \perp \mathbf{x}_0} \ \max_{\mathbf{v} \in \mathbb{R}^m} \ - s\rho_{init} - \mathbf{x}_{init}^{\mathsf{T}}\mathbf{w} + \lambda f(s\mathbf{x}_0 + \mathbf{w}) - \|\mathbf{v}\| \|\mathbf{h}^{\mathsf{T}}\mathbf{w} + |\mathbf{v}|^{\mathsf{T}}(| \ s\mathbf{q} + \|\mathbf{w}\|\mathbf{g} \ | - |\mathbf{q}|), \tag{46}$$

By fixing the norm of $\mathbf{v}$ and optimizing over its direction, the optimization problem (46) can be reduced to the following:

$$\min_{s \in \mathbb{R}, \ \mathbf{w} \perp \mathbf{x}_0} \quad - s\rho_{\text{init}} - \mathbf{x}_{\text{init}}^{\mathsf{T}}\mathbf{w} + \lambda f(s\mathbf{x}_0 + \mathbf{w})$$
$$\text{subject to:} \quad \mathbf{h}^{\mathsf{T}}\mathbf{w} \geq \|\{| \ s\mathbf{q} + \|\mathbf{w}\|\mathbf{g} \ | - |\mathbf{q}|\}_+\| , \tag{47}$$

where, for a vector $\mathbf{c}$, we let $\{\mathbf{c}\}_+$ denote the component-wise positive part function, with $i^{\text{th}}$ entry equal to $\max(0, \mathbf{c}_i)$. Next, note that $\mathbf{q}$ and $\mathbf{g}$ are independent vectors in $\mathbb{R}^m$ with i.i.d. standard normal entries. We introduce the function $\mathbf{c_d}(s, r)$ as follows:

**Definition 5.** *The function* $\mathbf{c_d} : \mathbb{R} \times \mathbb{R}_+ \to \mathbb{R}$ *is defined as,*

$$\mathbf{c_d}(s, r) = \mathbb{E}_{\mathbf{X}_1, \mathbf{X}_2}[\{|s\mathbf{X}_1 + r\mathbf{X}_2| - |\mathbf{X}_1|\}_+^2] , \tag{48}$$

*where* $\mathbf{X}_1, \mathbf{X}_2 \overset{i.i.d.}{\sim} \mathcal{N}(0, 1)$.

The next result creates a connection between the norm of the vector $\{|s\mathbf{q} + \|\mathbf{w}\|\mathbf{g}| - |\mathbf{q}|\}_+$, which appears in the constraint of (47), and the function $\mathbf{c_d}$.

**Lemma 6.** $\frac{1}{m}\|\{| \ s\mathbf{q} + \|\mathbf{w}\|\mathbf{g} \ | - |\mathbf{q}|\}_+\|^2 \xrightarrow{\mathbb{P}} \mathbf{c_d}(s, \|\mathbf{w}\|)$*, when $m \to \infty$.*

*Proof.* Define the vector $\mathbf{u} \in \mathbb{R}_+^m$ as,

$$\mathbf{u} := \{| \ s\mathbf{q} + \|\mathbf{w}\|\mathbf{g} \ | - |\mathbf{q}|\}_+ . \tag{49}$$

The entries of $\mathbf{u}$ are i.i.d and $\mathbb{E}[u_i^2] = \mathbf{c_d}(s, \|\mathbf{w}\|)$, for $1 \leq i \leq m$. Therefore, the weak law of large number gives the following:

$$\frac{1}{m}\|\mathbf{u}\|^2 = \frac{1}{m}\sum_{i=1}^m u_i^2 \xrightarrow{\mathbb{P}} \mathbb{E}[u_i^2] = \mathbf{c_d}(s, \|\mathbf{w}\|) . \tag{50}$$

$\square$

To conclude the proof of Lemma 4, we exploit the result of Lemma 6 to replace $\|\{|s\mathbf{q} + \|\mathbf{w}\|\mathbf{g}| - |\mathbf{q}|\}_+\|$ in (47), which gives us the following optimization:

$$
\begin{aligned}
\min_{s\in\mathbb{R}} \min_{\mathbf{w}\in\mathbb{R}^n, \mathbf{w}\perp\mathbf{x}_0} \quad & -\mathbf{x}_{\text{init}}^{\mathsf{T}}(s\mathbf{x}_0 + \mathbf{w}) + \lambda f(s\mathbf{x}_0 + \mathbf{w}) \\
\text{subject to:} \quad & \mathbf{h}^{\mathsf{T}}\mathbf{w} \geq \sqrt{m\,\mathbf{c_d}(s, \|\mathbf{w}\|)}\;.
\end{aligned}
\tag{51}
$$

Due to lack of space, we are not going through the technical details of obtaining the convergence result in (51). The point-wise convergence of the objective functions, for fixed values of $s$ and $\|\mathbf{w}\|$, follows from Lemma 6. To show the uniform convergence, we appeal to the convexity of the objective function. The corresponding convergence of the optimal cost follows from the uniform convergence.

The following lemma gives an explicit formula for the function $\mathbf{c_d}$ in terms of its two input arguments.

**Lemma 7.**

$$
\mathbf{c_d}(s,t) = \frac{1}{\pi}[((1+s)^2 + t^2)\,atan(\frac{t}{1+s})\; + ((1-s)^2 + t^2)\,atan(\frac{t}{1-s}) - 2t].
\tag{52}
$$

*Proof.*

$$
\mathbf{c_d}(s,t) = \mathbb{E}[\{|s\mathbf{X}_1 + t\mathbf{X}_2| - |\mathbf{X}_1|\}_+^2]
\tag{53}
$$

$$
= \frac{1}{\pi}\int_0^\infty e^{-x_1^2/2}\int_{\frac{1-s}{t}x_1}^\infty e^{-x_2^2/2}(tx_2 - (1-s)x_1)^2\; dx_2\; dx_1
\tag{54}
$$

$$
+ \frac{1}{\pi}\int_0^\infty e^{-x_1^2/2}\int_{-\infty}^{-\frac{1+s}{t}x_1} e^{-x_2^2/2}(tx_2 + (1+s)x_1)^2\; dx_2\; dx_1\;,
$$

where, due to symmetry, we have computed the expectation only for $\mathbf{X}_1 > 0$ and multiplied the result by two. In order to compute the integral change of variable and use the polar coordinates $(r,\theta)$ in the $x_1x_2$-plane. Recall that we have $x_1 = r\cos(\theta)$, and $x_2 = r\sin(\theta)$. Applying the change of variables would result the following:

$$
\mathbf{c_d}(s,t) = \frac{1}{\pi}\int_{\text{atan}(\frac{1-s}{t})}^{\pi/2}\int_0^\infty r^3 e^{-r^2/2}(t\sin(\theta) - (1-s)\cos(\theta))^2\; dr\; d\theta
\tag{55}
$$

$$
+ \frac{1}{\pi}\int_{\text{atan}(\frac{1+s}{t})}^{\pi/2}\int_0^\infty r^3 e^{-r^2/2}(t\sin(\theta) - (1+s)\cos(\theta))^2\; dr\; d\theta
$$

$$
= \frac{2}{\pi}\int_{\text{atan}(\frac{1-s}{t})}^{\pi/2}(t\sin(\theta) - (1-s)\cos(\theta))^2\; d\theta + \frac{2}{\pi}\int_{\text{atan}(\frac{1+s}{t})}^{\pi/2}(t\sin(\theta) - (1+s)\cos(\theta))^2\; d\theta
\tag{56}
$$

$$
= \frac{2}{\pi}[\frac{t^2 + (1-s)^2}{2}\text{atan}(\frac{t}{1-s}) - \frac{t(1-s)}{2}] + \frac{2}{\pi}[\frac{t^2 + (1+s)^2}{2}\text{atan}(\frac{t}{1+s}) - \frac{t(1+s)}{2}]
\tag{57}
$$

$$
= \frac{1}{\pi}[((1+s)^2 + t^2)\,\text{atan}(\frac{t}{1+s})\; + ((1-s)^2 + t^2)\,\text{atan}(\frac{t}{1-s}) - 2t]\;.
\tag{58}
$$

We derived (56) using the fact that $\int_0^\infty r^3 e^{-r^2/2}dr = 2$. Computing each of the integrals with respect to $\theta$ would result in (57), and the final result in (58) is derived by simplifying the terms in (57).

$\square$

## 5.4 Proof of Theorem 2

We start from the equivalent optimization derived as the result of Lemma 4, defined as,

$$\min_{s\in\mathbb{R}} \min_{\mathbf{w}\in\mathbb{R}^n,\mathbf{w}\perp\mathbf{x}_0} \quad \lambda f(s\mathbf{x}_0 + \mathbf{w}) - s\rho_{\text{init}} - \mathbf{x}_{\text{init}}^\mathsf{T}\mathbf{w} \tag{59}$$

$$\text{subject to:} \quad \mathbf{h}^\mathsf{T}\mathbf{w} \geq \sqrt{m\,\mathbf{c_d}(s,||\mathbf{w}||)}\,. \tag{60}$$

One key idea to analyze the optimization is to replace $f(s\mathbf{x}_0 + \mathbf{w})$ with its first-order linear approximation around the point $\mathbf{x}_0$. Let $\hat{f}$ denotes the approximation function,

$$\hat{f}(\mathbf{x}) = f(\mathbf{x}_0) + \max_{\mathbf{v}\in\lambda\partial f(\mathbf{x}_0)} \mathbf{v}^T(\mathbf{x} - \mathbf{x}_0)\,. \tag{61}$$

Here, $\partial f(\mathbf{x}_0)$ denotes the sub-differential of $f(\cdot)$ at point $\mathbf{x}_0$ which is well-defined for convex functions and is a compact and convex set. Replacing $f(\cdot)$ with its approximation, enables us to precisely analyze the conditions for perfect signal recovery in the equivalent optimization (60) which determines the precise phase transition in the regularized PhaseMax optimization. This approximation is tight when the norm of the error approaches zero (which occurs in perfect recovery). We refer the interested reader to [31] for more details.

Therefore, for the rest of this section, we will analyze the following optimization,

$$\min_{\substack{s\in\mathbb{R}\\\mathbf{w}\in\mathbb{R}^n,\mathbf{w}\perp\mathbf{x}_0}} \max_{\mathbf{v}\in\lambda\partial f(\mathbf{x}_0)} \quad f(\mathbf{x}_0) + \lambda\mathbf{v}^T(s\mathbf{x}_0 + \mathbf{w} - \mathbf{x}_0) - s\rho_{\text{init}} - \mathbf{x}_{\text{init}}^\mathsf{T}\mathbf{w}$$

$$\text{subject to:} \quad \mathbf{h}^\mathsf{T}\mathbf{w} \geq \sqrt{m\,\mathbf{c_d}(s,||\mathbf{w}||)}\,, \tag{62}$$

Next, we use the dual variable $\beta$ to rewrite (62) as

$$\min_{\substack{s\in\mathbb{R}\\\mathbf{w}\in\mathbb{R}^n,\mathbf{w}\perp\mathbf{x}_0}} \max_{\substack{\mathbf{v}\in\lambda\partial f(\mathbf{x}_0)\\\beta\geq 0}} f(\mathbf{x}_0) + \lambda\mathbf{v}^T(s\mathbf{x}_0 + \mathbf{w} - \mathbf{x}_0) - s\rho_{\text{init}} - \mathbf{x}_{\text{init}}^\mathsf{T}\mathbf{w} - \frac{\beta}{\sqrt{n}}\mathbf{h}^\mathsf{T}\mathbf{w} + \frac{\beta}{\sqrt{n}}\sqrt{m\,\mathbf{c_d}(s,||\mathbf{w}||)}\,. \tag{63}$$

In the next step, we would like to switch the minimization over $\mathbf{w}$ with the maximization over $\beta$. But since the objective function is not convex with respect to $\mathbf{w}$, such an exchange wouldn't be a direct result of the Sion's min-max theorem. However, note that the initial optimization satisfies the conditions of the Sion's min-max theorem. In the asymptotic settings, using the same techniques as in [39] (see section A.2.4 in the appendix of the paper), one can show that changing the order of min and max does not change the solution of the optimization problem.

Hence, we are now able to first do the minimization over $\mathbf{w}$. To do this, we define $r := ||\mathbf{w}||$ and buy fixing $r$ we are computing the minimization with respect to the direction of $\mathbf{w}$. The following optimization is the result of minimization over the direction of $\mathbf{w}$:

$$\min_{\substack{s\in\mathbb{R}\\r\geq 0}} \max_{\substack{\mathbf{v}\in\lambda\partial f(\mathbf{x}_0)\\\beta\geq 0}} (1-s)(\rho_{\text{init}} - \mathbf{v}^\mathsf{T}\mathbf{x}_0) - r\cdot\|\mathbf{P}^\perp(\lambda\mathbf{v} - \mathbf{x}_{\text{init}} - \frac{\beta}{\sqrt{n}}\mathbf{h})\| + \beta\sqrt{\alpha\,\mathbf{c_d}(s,r)}\,, \tag{64}$$

where as defined in Section 3, $\alpha = \frac{m}{n}$ is the oversampling ratio, and $\mathbf{P}^\perp = \mathbf{I} - \mathbf{x}_0\mathbf{x}_0^\mathsf{T}$ is the projection to the orthogonal subspace of $\mathbf{x}_0$.

Up to this point, the result is valid for every convex function $f(\cdot)$. But in order to continue our analysis, we need the following lemma which restricts us to a specific class of functions, i.e., the class of absolutely scalable functions.

**Lemma 8.** *Let* $f : \mathbb{R}^n \to \mathbb{R}$ *be a convex function such that for all* $\mathbf{x} \in \mathbb{R}$ *and* $\alpha \geq 0$, $f(\alpha\,\mathbf{x}) = \alpha\,f(\mathbf{x})$. *Then, for all* $\mathbf{v} \in \partial f(\mathbf{x})$,

$$\mathbf{v}^\mathsf{T}\mathbf{x} = f(\mathbf{x})\,,$$

*where* $\partial f(\mathbf{x})$ *is the set of sub-differentials of function* $f$ *at point* $\mathbf{x}$.

*Proof.* Since $f$ is convex, for all $\mathbf{v} \in \partial f(\mathbf{x})$ and any $\epsilon < 1$, we have

$$(1-\epsilon)\,f(\mathbf{x}) = f((1-\epsilon)\mathbf{x}) \geq f(\mathbf{x}) - \epsilon\mathbf{v}^\mathsf{T}\mathbf{x}\,. \tag{65}$$

Thus, $\epsilon\,f(\mathbf{x}) \leq \epsilon\mathbf{v}^\mathsf{T}\mathbf{x}$. Choosing $\epsilon_1 = 1/2$ and $\epsilon_2 = -1/2$ yields $\mathbf{v}^\mathsf{T}\mathbf{x} = f(\mathbf{x})$ which concludes the proof. $\qquad\square$

If we apply Lemma 8 to the objective function in (64), we can replace $\mathbf{v}^\mathsf{T}\mathbf{x}_0$ with $f(\mathbf{x}_0)$ for all $\mathbf{v} \in \partial f(\mathbf{x}_0)$, which gives the following optimization,

$$\min_{\substack{s\in\mathbb{R}\\r\geq 0}} \max_{\beta\geq 0} \; -(1-s)L(\mathbf{x}_0) - r \cdot \min_{\mathbf{v}\in\lambda\mathbf{P}^\perp\partial L(\mathbf{x}_0)} \left\| \mathbf{v} - \frac{\beta}{\sqrt{n}}\mathbf{h} \right\| + \beta\sqrt{\alpha}\,\mathbf{c_d}(s,r) \,. \tag{66}$$

Recall that $(1-s)$ and $r \geq 0$ respectively represent the norm of the error in the direction of $\mathbf{x}_0$ and its orthogonal complement. Therefore, the perfect recovery in our optimization corresponds to the case where the optimizers are $r^\star = 0$ and $s^\star = 1$, and we are interested in the phase transition ratio $\alpha^\star$ for which this happens.

We use the following approximation of the objective function near the point $(r,s) = (0,1)$, which was introduced earlier in (31),

$$\mathbf{c_d}(s,r) = \frac{1}{\pi}\left[((1-s)^2 + r^2)\mathrm{atan}(\frac{r}{1-s}) - r(1-s)\right] . \tag{67}$$

We idefine the new variable $t := \frac{r}{1-s}$ and rewrite the optimization in terms of $t$ and $s$. One can show from (67) that as $r \downarrow 0$ and $s \uparrow 1$, the value of $\mathbf{c_d}(s,r)$ will only depends on the ratio $t$.

$$\min_{\substack{s\in\mathbb{R}\\t\geq 0}} \max_{\beta\geq 0} \; \Psi(s,t,\beta) = -(1-s)L(\mathbf{x}_0) - t(1-s)\cdot\mathrm{dist}_{\lambda\partial L^\perp(\mathbf{x}_0)}(\frac{\beta}{\sqrt{n}}\mathbf{h}) + \beta(1-s)\sqrt{\alpha((1+t^2)\mathrm{atan}(t) - t)}\,,$$
$$\tag{68}$$

where $\partial L^\perp(\mathbf{x}_0) = \mathbf{P}^\perp\partial L(\mathbf{x}_0)$ and $\mathrm{dist}_\mathcal{S}(\mathbf{x})$ is the distance function defined in Definition 1. Since we have a convex-concave objective function over three scalars, we can write the first order optimality conditions for the solutions to (68),

$$\begin{cases} \frac{\partial}{\partial\beta}\Psi(s,t,\beta)\big|_{(s^\star=1,t^\star,\beta^\star)} = 0 \\ \frac{\partial}{\partial s}\Psi(s,t,\beta)\big|_{(s^\star=1,t^\star,\beta^\star)} = 0 \\ \frac{\partial}{\partial t}\Psi(s,t,\beta)\big|_{(s^\star=1,t^\star,\beta^\star)} = 0 \end{cases} . \tag{69}$$

We would like to find the conditions (on $\alpha$) under which the solution to (69) happens at $s^\star = 1$. Therefore, we aim to solve the system of nonlinear equations (69), for three unknowns $t, \beta$ and $\delta$.

These equations can be written in the following form,

$$-t \cdot \frac{\frac{\beta}{n}\|\mathbf{h}\|^2 - \frac{h^\mathsf{T}}{\sqrt{n}}\Pi_{\partial L^\perp(\mathbf{x}_0)}(\frac{\beta}{\sqrt{n}}\mathbf{h})}{\mathrm{dist}_{\lambda\partial L^\perp(\mathbf{x}_0)}(\frac{\beta}{\sqrt{n}}\mathbf{h})} + \sqrt{\alpha((1+t^2)\mathrm{atan}(t) - t)} = 0$$

$$L(\mathbf{x}_0) + t\cdot\mathrm{dist}_{\lambda\partial L^\perp(\mathbf{x}_0)}(\frac{\beta}{\sqrt{n}}\mathbf{h}) - \beta\sqrt{\alpha((1+t^2)\mathrm{atan}(t) - t)} = 0$$

$$-\mathrm{dist}_{\lambda\partial L^\perp(\mathbf{x}_0)}(\frac{\beta}{\sqrt{n}}\mathbf{h}) + \frac{\beta\,t\,\mathrm{atan}(t)\sqrt{\alpha}}{\sqrt{\alpha((1+t^2)\mathrm{atan}(t) - t)}} = 0 \tag{70}$$

Next, we are going to exploit the conditions of assumption 1 (see Section 3.2). Using theorem 5.2.2. in [44], both the functions $\mathrm{dist}_{\lambda\partial L^\perp(\mathbf{x}_0)}(\frac{\beta}{\sqrt{n}}\mathbf{h})$ and $\frac{h^\mathsf{T}}{\sqrt{n}}\Pi_{\partial L^\perp(\mathbf{x}_0)}(\frac{\beta}{\sqrt{n}}\mathbf{h})$ converge point-wise to their expected value. Moreover, from assumption 1, we know that both $\mathbb{E}[\mathrm{dist}_{\lambda\partial L^\perp}(\mathbf{x}_0)(\frac{\beta}{\sqrt{n}}\mathbf{h})]$ and $\mathbb{E}[\frac{h^\mathsf{T}}{\sqrt{n}}\Pi_{\partial L(\mathbf{x}_0)}(\frac{\beta}{\sqrt{n}}\mathbf{h})]$ converge uniformly to $G_\lambda(\beta)$ and $\beta - F_\lambda(\beta)$, respectively. Therefore, using the same arguments as in [39], we can replace $\mathrm{dist}_{\lambda\partial L^\perp(\mathbf{x}_0)}(\frac{\beta}{\sqrt{n}}\mathbf{h})$ with $F_\lambda(\beta)$ in the optimization (68), and then apply the result of theorem 7.17 in [34], we can show that $F'_\lambda(\beta) = G_\lambda(\beta)\,G'_\lambda(\beta)$.

Therefore, we are able to use the functions $F_\lambda$, and $G_\lambda$ to rewrite the system of non-linear equations (70):

$$- t \cdot \frac{F_\lambda(\beta)}{G_\lambda(\beta)} + \sqrt{\alpha((1+t^2)\mathrm{atan}(t) - t)} = 0$$

$$L(\mathbf{x}_0) + t \cdot G_\lambda(\beta) - \beta\sqrt{\alpha((1+t^2)\mathrm{atan}(t) - t)} = 0$$

$$- G_\lambda(\beta) + \frac{\beta\, t\, \mathrm{atan}(t)\sqrt{\alpha}}{\sqrt{\alpha((1+t^2)\mathrm{atan}(t) - t)}} = 0 \tag{71}$$

By combining the first and third equations, we will get

$$t = \tan\left(\frac{\pi}{\alpha\beta} F_\lambda(\beta)\right) \tag{72}$$

Finally, using (72) in (71) reduces the number of equations to 2, and yields the following system of non-linear equations.

$$\begin{cases} G_\lambda(\beta)\, l = \tan(\frac{\pi}{\alpha\beta}F_\lambda(\beta))\left(G_\lambda^2(\beta) - \beta F_\lambda(\beta)\right), \\ \tan(\frac{\pi}{\alpha\beta}F_\lambda(\beta))\left(G_\lambda(\beta) - \frac{\pi l}{\alpha\beta}F_\lambda(\beta)\right) = \frac{\pi}{\alpha\beta}F_\lambda(\beta)\, G_\lambda(\beta), \end{cases} \tag{73}$$

This concludes the proof.

## 5.5 Computing the statistical dimension for the $\ell_1$ regularization

In this section we bound the statistical dimension of the descent cone of the objective function of (2), where $f(\cdot) = ||\cdot||_1$ is used for regularization. We assume that the underlying signal, $\mathbf{x}_0$, is $k$-sparse and define function $L_\lambda : \mathbb{R}^n \to \mathbb{R}$ as follows,

$$L_\lambda(\mathbf{x}) := -\mathbf{x}_{\mathrm{init}}^\mathsf{T}\mathbf{x} + \lambda||\mathbf{x}||_1, \tag{74}$$

In order to derive an upper bound for the statistical dimension $d(T_{L_\lambda}(\mathbf{x}_0))$, we first introduce another summary parameter for convex sets called the Gaussian width.

**Definition 6.** *(Gaussian width) [44] The Gaussian width of a subset $\mathcal{T} \subset \mathbb{R}^n$ is defined as,*

$$\omega(\mathcal{T}) = \mathbb{E}\sup_{\mathbf{x}\in\mathcal{T}}\langle\mathbf{x}, \mathbf{g}\rangle, \quad \text{where } \mathbf{g} \sim \mathcal{N}(\mathbf{0}, \mathbf{I}). \tag{75}$$

We use the following proposition which indicates the relationship between the Gaussian width and statistical dimension of a convex cone.

**Proposition 1.** *(Proposition 10.2 in [1]) Let $\mathcal{C} \subset \mathbb{R}^n$ be a convex cone. Then*

$$\omega^2(\mathcal{C} \cap \mathbb{S}^{n-1}) \le d(\mathcal{C}) \le \omega^2(\mathcal{C} \cap \mathbb{S}^{n-1}) + 1, \tag{76}$$

*where $\mathbb{S}^{n-1} \subset \mathbb{R}^n$ is the unit sphere.*

The Proposition 1 shows that in order to bound the statistical dimension of a convex cone, we need to bound the squared Gaussian width of that cone. Hence, in the remaining we will bound the squared Gaussian width. We shall briefly review some known properties of Gaussian width of convex cones.

### 5.5.1 Some properties of Gaussian width

The Gaussian width is one of the intrinsic volumes of a body studied in combinatorial geometry. It is invariant under translation and unitary transformation and has deep connections to convex geometry. While discussing all the properties of Gaussian width is beyond the scope of this paper, we refer the interested reader to [33, 42, 44] and references therein.

Inspired by [38, 11], here we bound the Gaussian width of a cone via the distance to its polar cone. Before stating the proposition, we recall the definition of the polar cone.

**Definition 7.** *(Polar cone) Let $\mathcal{C} \subset \mathbb{R}^n$ be a non-empty convex cone. The polar cone of $\mathcal{C}$, denoted by $\mathcal{C}^\star$, is defined as follows,*

$$\mathcal{C}^\star = \{\mathbf{z} \in \mathbb{R}^n : \langle \mathbf{z}, \mathbf{x} \rangle \leq 0 \ \textit{for all } \mathbf{x} \in \mathcal{C}\} . \tag{77}$$

The following proposition establishes a connection between the Gaussian width of the cone $\mathcal{C}$ and its polar cone $\mathcal{C}^\star$:

**Proposition 2.** *(Proposition 3.6 in [11]). Let $\mathcal{C}$ be any non-empty convex cone in $\mathbb{R}^n$, and let $\mathbf{g} \sim \mathcal{N}(0, \mathbf{I})$ be a random Gaussian vector. Then we have the following bound:*

$$\omega(\mathcal{C} \cap \mathbb{S}^{n-1}) \leq \mathbb{E}_{\mathbf{g}}[dist(\mathbf{g}, \mathcal{C}^\star)] , \tag{78}$$

*where the $dist(\cdot, \cdot)$ function here denotes the Euclidean distance between a point and a set.*

Applying Jensen's inequality will result in the following,

$$\omega^2(\mathcal{C} \cap \mathbb{S}^{n-1}) \leq \mathbb{E}_{\mathbf{g}}[\mathrm{dist}^2(\mathbf{g}, \mathcal{C}^\star)] . \tag{79}$$

This is very useful in bounding the Gaussian width of the descent cone of a convex function due to the following lemma:

**Lemma 9.** *[32] For a convex function $f : \mathbb{R}^n \to \mathbb{R}$,*

$$(T_f(\mathbf{x}))^\star = cone(\partial f(\mathbf{x})) , \tag{80}$$

*where $\partial f(\mathbf{x})$ is the sub-differential set of function $f$ at point $\mathbf{x}$.*

The polar cone of $T_f(\mathbf{x})$ is also called the *normal cone*, $N_f(\mathbf{x})$, at point $\mathbf{x}$. Exploiting the above results, we can bound $\omega^2(T_L(\mathbf{x}_0) \cap \mathbb{S}^{n-1})$ in terms of the squared distance to the normal cone at point $\mathbf{x}_0$, i.e., we have the followings,

$$d(T_{L_\lambda}(\mathbf{x}_0)) \leq \omega^2(T_{L_\lambda}(\mathbf{x}_0) \cap \mathbb{S}^{n-1}) + 1 \leq \mathbb{E}_g[\mathrm{dist}^2(\mathbf{g}, \mathrm{cone}(\partial L_\lambda(\mathbf{x}_0))] + 1 . \tag{81}$$

For simplicity in the remaining formulations we omit the sub-script $\lambda$ and denote the objective function by $L$. Let $\Delta$ denote the set of coordinates where $\mathbf{x}_0$ is non-zero. The sub-differential set of the function $L$ (defined in (74)) can be formally characterized as follows,

$$\partial L(\mathbf{x}_0) = \{-\mathbf{x}_{\mathrm{init}} + \frac{\lambda}{\sqrt{k}} \mathbf{v} : \mathbf{v} \in \mathbb{R}^n \ \text{s.t. } \mathbf{v}[i] = \mathrm{sign}(\mathbf{x}_0[i]) \text{ for } i \in \Delta, \ |\mathbf{v}[i]| \leq 1 \text{ for } i \in \Delta^c\}. \tag{82}$$

Here $\Delta^c := [n] - \Delta$ represents the zero entries of $\mathbf{x}_0$. Without the loss of generality, we are going to assume that the first $k$ entries of $\mathbf{x}_0$ are non-zero, while the rest are zero. Then, cone of the sub-differential can be written as

$$\mathrm{cone}(\partial f(\mathbf{x})) = \{\beta \cdot (-\mathbf{x}_{\mathrm{init}} + \frac{\lambda}{\sqrt{k}} \mathbf{v}) : \mathbf{v}[1:k] = \mathbf{1}, \ \|\mathbf{v}[k+1:n]\|_\infty \leq 1, \ \beta \geq 0\}. \tag{83}$$

The squared distance to the normal cone can be formulated as the following optimization:

$$\mathrm{dist}^2(\mathbf{g}, N_L(\mathbf{x}_0)) = \min_{t \geq 0} (\sum_{i \in \Delta} (\mathbf{g}[i] + t\mathbf{x}_{\mathrm{init}}[i] - t\lambda\mathrm{sign}(\mathbf{x}_0[i]))^2 \tag{84}$$
$$+ \min_{j \in \Delta^c, |u_j| < t} (\mathbf{g}[j] + t\mathbf{x}_{\mathrm{init}}[j] - \lambda u_j)^2)$$

Define $\mathbf{z} := \mathbf{z}(t) = \mathbf{g} + t\mathbf{x}_{\mathrm{init}}$. We can rewrite the equation (84) as follows,

$$\text{dist}^2(\mathbf{g}, N_L(\mathbf{x}_0)) = \min_{t \geq 0} (\sum_{i \in \Delta} (\mathbf{z}[i] - t\lambda \text{sign}(\mathbf{x}_0[i]))^2 \tag{85}$$

$$+ \sum_{j \in \Delta^c} \text{shrink}(\mathbf{z}[j], t\lambda)^2) \,,$$

where the function $\text{shrink}(\cdot, \cdot)$ is defined as, $\text{shrink}(x, T) = \begin{cases} x + T \,, & x < -T \\ x \,, & -T \leq x \leq T \\ x - T \,, & x > T \end{cases}$ . This function

is known as $\ell_1$-shrinkage function and is used in sparse denoising. Taking the expectation with respect to $\mathbf{g}$ will provide us with the quantity we would like to bound. We are bounding the expectation of the squared distance by bounding each of the two terms of the sum. For the first term, we have:

$$\mathbb{E}[\sum_{i \in \Delta} (\mathbf{z}[i] - t\lambda \text{sign}(\mathbf{x}_0[i]))^2] = k + t^2(\lambda^2 k - ||\mathbf{x}_{\text{init}}^\Delta||^2) - 2t\lambda \text{sign}(\mathbf{x}_0)^T \mathbf{x}_{\text{init}}^\Delta \tag{86}$$

Bounding the expectation of the second term in (85) is more involved. Using the same techniques as Appendix C in [11], we can show the following:

$$\mathbb{E}[\sum_{j \in \Delta^c} \text{shrink}^2(\mathbf{z}[j], t\lambda)] \leq \frac{2(n-k)\sigma^3}{\sqrt{2\pi}t\lambda} exp(\frac{-t^2\lambda^2}{2\sigma^2}) \,, \tag{87}$$

where $\sigma^2 := 1 + t^2 \frac{||\mathbf{x}_{\text{init}}^\Delta||^2}{n-k}$.

Using the result of equations (86) and (87), we can see that when $\lambda > \frac{c}{\sqrt{k}}$, then by choosing $t = \frac{\sqrt{2\log \frac{n}{k}}}{\lambda}$, both of the terms in the sum are bounded by $Ck\log \frac{n}{k}$, where $c$ and $C$ are constants that are independent of the problem's parameters.

Therefore, when $\lambda > \frac{c}{\sqrt{k}}$ (where $k$ is the number of non-zero entries), the statistical dimension of $T_{L_\lambda}(\mathbf{x}_0)$ is bounded by $Ck\log \frac{n}{k}$. Using the result of Theorem 1, we can conclude that for the sparse phase retrieval problem the required sample complexity of the regularized PhaseMax is $\mathcal{O}(k\log \frac{n}{k})$. This indicates that regularized PhaseMax is order-wise optimal.