[Reviews · NeurIPS 2018]

Reviewer 1



This paper presents a sharp asymptotic analysis of the regularized phase-max estimator for structured phase retrieval (PR) problems. The phase-max algorithm is a recently proposed Linear Programming based estimator for PR. The finite sample as well as the sharp asymptotic performance of phase-max for unstructured phase retrieval was well understood from prior work. The authors propose a natural modification by penalizing the usual phase-max objective with a regularizer $f$. For structured PR, prior work showed that for a $k$-sparse signal vector, L1-penalized phase-max recovers the signal with $m = O(klog(n/k))$ samples given a good enough initialization. The primary contribution of this work is (i) extending the result to absolutely scalable regularizers which allows for other kinds of structural assumptions on the signal (ii) complementing the finite sample with a high dimensional asymptotic result giving an explicit formula for the asymptotic sampling ratio $\alpha = m/n$ required to achieve exact recovery in terms of the quality of the initialization and the oversampling ratio $\delta = d/m$ where $d$ is the effective degrees of freedom of the signal vector. The main tool used to establish this result is the Convex Gaussian Min-Max (CGMT) framework which is a well-established technique from prior work. The quality and clarity of the work can be improved. Several of the statements of the theorems and lemmas are not very precise and some of the arguments are non-rigorous. However, the results are definitely interesting and plausible (see detailed comments below). They are also supported by simulations. Section 4.5 is clearly incomplete and ends abruptly. Citations in paragraph 509-515 are informal notes. To the best of my knowledge, the formula for the phase transition presented by the authors was not known before and is a significant and original contribution. While CGMT is a well-established technique, applying it to this problem seems to require some original and interesting problem specific analysis. Detailed Comments about proofs: Lines 409-412: The CGMT lemma requires additional assumptions which are omitted and not verified in the proof. The compactness assumption on $S_w$ and $S_u$ is stated but not verified when invoking this lemma in 545-547. This usually requires some technical arguments. Lemma 3, Eq 23: The claimed $n \rightarrow \infty$ limit depends on $n$. Lines 434-436: Approximation of $f$ by the linear underestimator is justified non-rigorously. Likewise in lines 471-475. Lines 480-482: Can the authors state which lemma/theorem of [Thrampoulidis et al, 2016] they are invoking to justify changing the order of min-max? I am unable to find such a result in that paper and verify their claim. Pages 17-18: To apply the CGMT theorem, the authors need to verify that the AO optimal value and the norm of the AO optimal solution converge in probability. It appears to me this has not been done rigorously. In particular the AO and PO have a distinctive feature that they depend on additional randomness $q$ apart from the usual randomness $A, g, h$. The authors don’t state how they handle this. CGMT is a general purpose theorem and most of the work involved is in verifying its technical conditions and it would nice to have these details in the appendix. Lines 198-201: I don’t think the citation [9] calculates the statistical dimension of the descent cone of this objective. I can see this was being done in the appendix section 4.5 which is incomplete. Other minor comments: Lines 98-100: It should be clarified that the paper by Soltanolkotabi [33] breaks the \emph{local} sample complexity barrier (sample complexity required for recovery when initialized closed enough to the truth) and not the global complexity barrier. Lines 133-138: The reference stated does not justify the optimality of the sample complexity. To my understanding, these references give the sample complexity required for certain estimators to succeed for linear measurements. It might be more appropriate to have a reference for an information theoretic lower bound/ identifiability lower bound in the specific context of phase retrieval since the title of the paper also claims optimality. Line 213: orthogonal projection of sub-diff .. into the orthogonal… : Remove first orthogonal. Line 216: Missing linebreak at Assumption 1. Figure 1(b): missing legend. Lines 237-241: Use \begin{remark} Line 187: Missing linebreak at Example 1. Lines 167-168: Has the claim about the function $R$ been proved anywhere in the paper? If it is a known result, it needs a citation. Lines 169-171: Is the requirement on $R(\delta)$ for all $n$? -------------------------------------------------------------------- POST-REBUTTAL: My primary concern regarding this paper was that some of statements and arguments appeared non-rigorous and imprecise to me. The authors addressed these issues in their rebuttal. Furthermore since I do think this result is interesting to the NIPS community, I am increasing my score to 7.

Reviewer 2



PhaseMax is a recently developed convex relaxation of the phase retrieval problem that avoids the need to perform computationally intractable lifting. In this work, the authors derive necessity and sufficiency conditions to solve regularized versions of phase max, i.e., phasemax plus a regularization penalty, for instance an L1 norm. In order to derive (slightly pessimistic) sufficiency conditions, the authors relate phasemax to another optimization problem that admits easier analysis. In order to derive (fairly sharp) necessity conditions, the authors analyze a linear approximation of the the equivalent optimization problem from before. Both conditions are written in terms of the quality of the initializer, that is how correlated the initialization is with the truth, and the amount of oversampling, written in terms of the intrinsic dimension of the signal. The paper studies an important subject, makes an interesting contribution, and is generally well written. I have a few comments, listed below, but I feel the paper is worth accepting. Comments: More information should be provided on how one chooses the initialization. With complex $x$ and without prior information, "Fundamental Limits of Weak Recovery with Applications to Phase Retrieval" demonstrated that one cannot come up with a good initialization when m/n<1-o(m/n). Is there a way to initialize $x$ with fewer measurements when one knows something about it? If not, the results in this paper are not that useful. How do the results presented in this work relate to those in [18]? Is [18] looser? Minor Issue: Authors should point readers to supplement to find proofs.

Reviewer 3



UPDATE AFTER READING THE AUTHORS' RESPONSE: Thanks for the clarification/explanation about the missing text, that is very helpful! Summary of the paper: This paper addresses the problem of phase retrieval of structured (e.g., sparse) signals. The authors propose a regularized variant of the PhaseMax convex programming formulation. For the special case of Gaussian random measurements, they give a thorough analysis, which shows that regularized PhaseMax succeeds with a near-optimal number of measurements, and also provides tight bounds on the location of the "phase transition" for successful recovery of the unknown signal. Overall, I think this is an interesting result, with strong technical content. I think it is significant, because previous attempts at using convex relaxations (e.g., PhaseLift) for sparse phase retrieval have run into serious obstacles. Given these difficulties, I think it is nice to have a positive result, even if it is in a somewhat artificial setting (with Gaussian random measurements) that may not be useful for practitioners. The proof uses standard tools from convex geometry, such as convex duality, descent cones and statistical dimension, which I know well; as well as a more recent technique, the "convex Gaussian min-max theorem," which I do not know so well. One question: Where is the proof of Lemma 1 (stated in Section 3.2.2)? I looked in the Supplementary Material, and it appears to be in Section 4.5, but the document ends abruptly, without finishing that line of argument? Also, note that some of the references on page 16 are missing.